# Net heterotrophy and carbonate dissolution in two subtropical seagrass meadows

Bryce R Van Dam[1,2], Christian Lopes[2], Christopher L Osburn[3], James W Fourqurean[2]

[1] Dept of Biological Sciences and Center for Coastal Oceans Research, Florida International University, 11200 SW 8th St, Miami FL 33199, USA
[2] Institute of Coastal Research, Helmholtz-Zentrum Geesthacht (HZG), Geesthacht, 21502, Germany
[3] Dept of Marine, Earth, and Atmospheric Sciences, North Carolina State University, 2800 Faucette Drive, Raleigh, North Carolina 27695, USA

*Correspondence to*: Bryce R Van Dam (vandam.bryce@gmail.com)

**Abstract.** The net ecosystem productivity (NEP) of two seagrass meadows within one of the largest seagrass ecosystems in the world, Florida Bay, was assessed using direct measurements over consecutive diel cycles during a short study in the Fall of 2018. We report significant differences between NEP determined by dissolved inorganic carbon ($NEP_{DIC}$) and by dissolved oxygen ($NEP_{DO}$), likely driven by differences in air-water gas exchange and contrasting responses to variations in light intensity. We also acknowledge the impact of advective exchange on metabolic calculations of NEP and net ecosystem calcification (NEC) using the 'open water' approach, and attempt to quantify this effect. In this first direct determination of $NEP_{DIC}$ in seagrasses, we found that both seagrass ecosystems were net heterotrophic, on average, despite large differences in seagrass net aboveground primary productivity. NEC was also negative, indicating that both sites were net dissolving carbonate minerals. We suggest that a combination of carbonate dissolution and respiration in sediments exceeded seagrass primary production and calcification, supporting our negative NEP and NEC measurements. However, given the limited spatial (two sites) and temporal (8 days) extent of this study, our results may not be representative of Florida Bay as a whole and may be season-specific. The results of this study highlight the need for better temporal resolution, accurate carbonate chemistry accounting, and an improved understanding of physical mixing processes in future seagrass metabolism studies.

## 1 Introduction

Seagrass ecosystems are often net autotrophic, producing more organic matter than they consume (Duarte et al, 2005; Barrón et al., 2006; Duarte et al, 2010; Unsworth et al., 2012; Long et al., 2015a; Ganguly et al., 2017; Perez et al., 2018). In terrestrial ecosystems, $CO_2$ uptake by photoautotrophs necessarily leads to an exchange of carbon from the atmosphere to the biosphere. However, such a net uptake of $CO_2$ by submerged seagrasses is attenuated as carbon produced or consumed by net ecosystem productivity (NEP) interacts with the carbonate buffering system and the processes of calcification and carbonate dissolution in the water, submerged sediments, and calcifying organisms. The impact of seagrass carbonate chemistry on

measurements of NEP is further obscured by physical processes at the air-water interface, which may cause temporal lags between NEP and air-water $CO_2$ exchange.

Calcification is an important process in many tropical and subtropical seagrass ecosystems (Mazarrasa et al. 2015) and has the net effect of consuming total alkalinity (TA) in excess of dissolved inorganic carbon (DIC), thereby decreasing pH and generating $CO_2$. Florida Bay is a well-studied seagrass-dominated ecosystem and is assumed to be net calcifying given the vast autochthonous sedimentary deposits of $CaCO_3$ that have accumulated in the bay in the last three millennia (Stockman et al., 1967; Bosence et al., 1985). While much of this $CaCO_3$ was produced by other photoautotrophic or non-photoautotrophic calcifiers (Frankovich and Zieman 1994), it is likely that some unknown fraction was also derived from calcification driven directly by the seagrasses (Enríquez et al., 2014), although the extent to which internal $CaCO_3$ formation occurs remains a debated topic. Existing measurements from Florida Bay show that net ecosystem calcification (NEC) can vary from positive to negative over diel cycles (Turk et al., 2015), and across gradients of seagrass productivity and substrate type (Yates and Halley 2006). The relative magnitudes of NEC and NEP in the context of the overall seagrass ecosystem carbon budget is unclear, and it is still uncertain which component of the ecosystem dominates net calcification (seagrasses, benthic invertebrates, macroalgae, etc.). Early assessments of seagrass NEC in Florida Bay relied on species-specific calcification rates that were up-scaled to the community or ecosystem level. These studies indicate that epiphytic calcification can dominate NEC (Frankovich and Zieman 1994), and that the physical transport of carbonate mud within the bay is likely significant (Bosence 1989). The physical transport of carbonate mud is important because it can allow $CaCO_3$ formation and destruction to become spatially decoupled, such that regions of net dissolution may exist within the larger context of a net calcifying Florida Bay. More recently, results from in-situ chambers have indicated that seagrass primary production can dominate short-term carbonate chemistry dynamics (Hendriks et al., 2014; Turk et al., 2015; Camp et al., 2016).

This biological $CO_2$ addition or removal causes non-linear changes in the marine carbonate system, further challenging direct measurements of seagrass ecosystem NEP. Hence, prior assessments of seagrass NEP were often made using dissolved oxygen production (DO) as a proxy for $CO_2$ fixation, necessitating the assumption of a photosynthetic quotient (PQ) relating $CO_2$ fixation to DO production. The assumption of a PQ value is made problematic by the carbonate system reactions discussed earlier, which affect $CO_2$ but not DO. While it is often assumed that PQ is approximately 1 (e.g., Duarte et al., 2010), prior measurements of $\Delta CO_2/\Delta DO$ in seagrass ecosystems show a wide range of values, from 0.3 to 6.8 (Ziegler and Benner 1998; Barrón et al., 2006; Turk et al., 2015). As a result, potential exists for a general disagreement between NEP assessed using measurements of carbon, and those using its $O_2$ proxy ($NEP_{DO}$). Hence, we identify a need for simultaneous measurements of pH, $O_2$, $pCO_2$, TA and dissolved inorganic carbon (DIC) when assessing seagrass ecosystem NEP and NEC, which may explain the divergence between $CO_2$- and $O_2$-based methods.

In addition to the importance of primary production in seagrass meadows as a source of energy to fuel coastal ecosystems, the net uptake of $CO_2$ from the overlying water could have other important impacts of the seascapes in which the seagrasses occur. High primary production drives large diel variations in pH within seagrass meadows (e.g. Hendriks et al., 2014; Turk et al., 2015; Camp et al., 2016; Challener et al., 2016), and it has been suggested that seagrass NEP may partially

buffer coastal ocean acidification (OA) by consuming $CO_2$, thereby creating refugia for calcifying organisms (Manzello et al., 2012; Unsworth et al., 2012; Hendriks et al., 2014; Koweek et al., 2018; Pacella et al., 2018). Seagrasses may also help to buffer local changes in pH by attenuating mangrove-derived fluxes of DIC (Buillon et al 2007). However, it remains unclear how NEP and NEC might interactively affect carbonate system buffering in regions where primary producer biomass and NEP

are limited by the availability of nutrients, like in the severely phosphorus-limited regions of Florida Bay (Fourqurean et al. 1992).

Prior studies of $NEP_{DO}$ in Florida Bay have suggested net autotrophy (Long et al., 2015a), yet others were unable to infer long-term $NEP_{DO}$ balance (Turk et al., 2015). Both of these estimates of $NEP_{DO}$ necessarily ignore any anaerobic catabolic biogeochemical processes that may cause $NEP_{DIC}$ to decrease, but do not affect $NEP_{DO}$. Rates of denitrification (Eyre and

10 Ferguson 2002) and sulfate reduction (Smith et al., 2004, Ruiz-Halpern et al., 2008) can be significant in seagrass soils, although rates may depend on specific seagrass morphology and physiological traits (Holmer et al., 2001). Additionally, despite the inferred net ecosystem autotrophy of seagrasses, $pCO_2$ is often found above (Millero et al., 2001) or near (Yates et al., 2007) equilibrium with the atmosphere throughout most of Florida Bay, suggesting the important role of NEC or anaerobic catabolic processes in generating excess $CO_2$.

In this study, we describe our direct measurements of $NEP_{DIC}$, $NEP_{DO}$, and NEC in two Florida Bay seagrass sites. We investigate variations in NEP and NEC across a seagrass productivity gradient, discuss differences between $NEP_{DIC}$ and $NEP_{DO}$, and suggest possible drivers of NEP and NEC.

## 2 Methods

### 2.1 Study Site

This study took place in one of the largest seagrass ecosystems in the world, Florida Bay (Figure 1), where we occupied two primary study sites which experience similar hydrologic and climatologic conditions yet differ substantially in community composition and biomass (Table 1). The choice of these sites allowed us to discern the effects of seagrass abundance and productivity on NEP and NEC that are independent of environmental setting. Both sites were dominated by the seagrass *Thalassia testudinum* in a phosphorus limited region (Fourqurean et al., 1992), have similar water depths (~2m), and

were approximately 0.5 - 1 km from land. However, these sites differed in important factors like seagrass above-ground biomass, nutrient content, morphology, as well as sediment depth, soil carbon (organic and inorganic), and soil nutrient content (Table 1). The potential for submarine groundwater discharge at these locations is low (Corbett et al., 1999). In addition to the two primary study sites, we collected time series data of DO and pH for an additional four Florida Coastal Everglades Long Term Ecological Research (FCE-LTER) sites in an effort to test whether the relationship between $NEP_{DO}$ and $NEP_{DIC}$ observed

in this study can be extended over larger areas of Florida Bay.

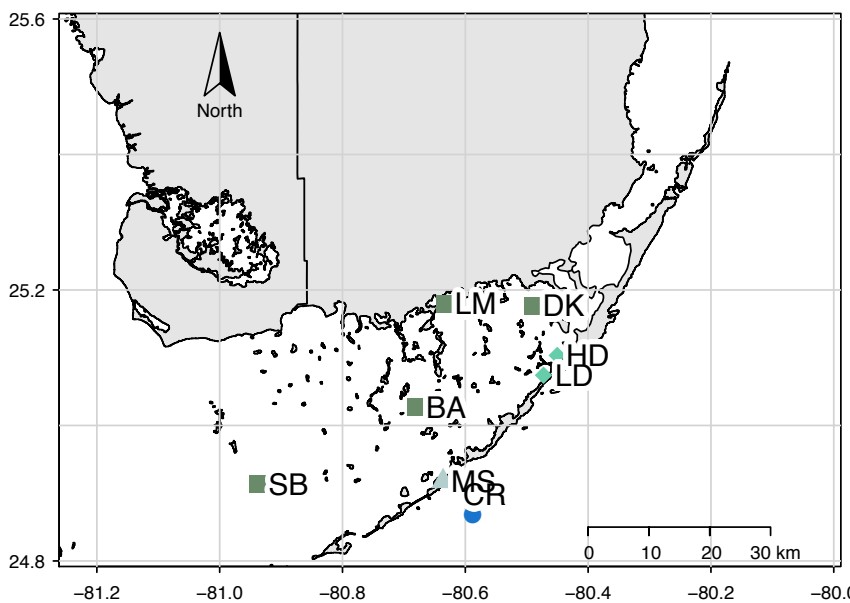

**Figure 1.** Site map, showing locations of the high- and low-density sites (HD and LD), meteorological stations used to derive $U_{10}$ and $pCO_2$ data (MS and CR, respectively). Additional FCE-LTER sites used in this study are shown as the dark green squares: Sprigger Bank (SB), Bob Allen (BA), Little Madeira (LM), and Duck Key (DK).

## 2.2 Sampling Campaigns

We quantified $NEP_{DO}$, $NEP_{DIC}$, and NEC at our high density-and low-density sites by measuring diel excursions in DO, DIC and TA, and applying corrections to account for factors like air-water gas exchange and variations in water depth and light intensity. This is essentially a modification of the 'free-water' approach to assessing NEP (Nixon et al., 1976; Odum and Hoskin 1958), where the total inventory of DIC or $O_2$ is monitored over time. A benefit of this approach over traditional chamber-based metabolism methods is that the container effect is avoided, which is known to result in under-estimations of benthic respiration, due to a dampening of turbulent sediment-water exchange (Hopkinson and Smith, 2007). This approach has a number of weakness, however, related both to the reliance on modelled air-water gas exchange, which is subject to a high degree of uncertainty (Upstill-Goddard 2006), and the assumption that the system is closed and does not exchange water or material with adjacent systems. Both of these assumptions may be broken in shallow seagrass meadows, where tides are minimal but wind-driven seiche can be important. Furthermore, the physics governing air-water gas exchange in these systems are very poorly understood, and while it is assumed that wind-driven turbulence is the dominant driver, other factors like convection (MacIntyre et al., 2010; Podgrajsek et al., 2014), bottom-driven turbulence (Ho et al., 2016; Raymond and Cole 2001), surfactant activity (McKenna and McGillis 2004; Lee and Saylor 2010), and chemical enhancement may at times play an equal or greater role (Smith 1985; Wanninkhof 1992).

During two sampling campaigns in late 2018, measurements were made over consecutive diel cycles for a total of 8 days. The first campaign lasted for ~4 days from Oct. 28 - Nov. 01, while the second campaign, also ~4 days, lasted from Nov. 25 – Nov. 29. Samples were taken 3 times per day during the first campaign (dawn, noon, and dusk), and 4 times per day during the second campaign (dawn, late morning, early afternoon, and dusk). During the first sampling campaign, water

samples were collected for the analysis of stable isotopic composition of DIC ($\delta^{13}C_{DIC}$), in an effort to constrain potential DIC sources. We applied Keeling plots to our isotopic data, where $1/nDIC$ is plotted against $\delta^{13}C_{DIC}$. In this approach, the y-intercept (as $1/nDIC$ approaches 0) indicates the $\delta^{13}C_{DIC}$ value as $nDIC$ approaches infinite concentration (e.g., as $1/nDIC$ approaches 0) and can be interpreted as an indicator of the $\delta^{13}C_{DIC}$ of the source of the DIC (Karlsson et al., 2007).

## 2.3 Discrete Measurements

At our primary study sites, water samples for total alkalinity (TA) and dissolved inorganic carbon (DIC) were collected with pre-rinsed borosilicate bottles at a depth of approximately 0.2 m. TA and DIC samples were preserved with a saturated solution of $HgCl_2$ and stored on ice until analysis (Dickson et al., 2007). Samples for $\delta^{13}C_{DIC}$ were taken at the same depth, filtered to 0.45µm, and preserved with $HgCl_2$. Calcite saturation state ($\Omega_{calcite}$) was calculated in CO2Sys (Lewis and Wallace 1998) from measured TA, DIC, salinity and temperature, using the $H_2CO_3$ dissociation constants of Mehrbach et al.

(1973) refit by Dickson and Millero (1987).

At each of our primary sites, small quadrats (n = 6, 10 cm × 20 cm) were randomly placed, at which aerial seagrass primary productivity ($g\,m^{-2}\,d^{-1}$) rates were determined using the leaf marking technique (Zieman et al. 1989). For this analysis, seagrass leaves were scraped of all epiphytes using a razor blade, rinsed, and dried at 65 °C until a constant weight. This dried seagrass material was then weighed as seagrass biomass. Dry samples were homogenized and ground to a fine powder using

a motorized mortar and pestle in preparation for tissue elemental content analysis (C,N,P). Powdered samples were analysed for total carbon (TC) and nitrogen content using a CHN analyser (Thermo Flash EA, 1112 series). Phosphorus content was determined by a dry-oxidation, acid hydrolysis extraction followed by a colorimetric analysis of phosphate concentration of the extract (Fourqurean and Zieman 1992). Elemental ratio is reported as mole:mole. Surface soils were collected using a 60 mL manual piston core following previously described methods for determining soil carbon content ($C_{org}$ and $C_{inorg}$)

(Fourqurean et al. 2012b).

## 2.4 Continuous Measurements

At each of our primary sites, we deployed a YSI EXO-2 water quality sonde which recorded water depth, sea surface temperature (SST, °C), sea surface salinity (SSS), and dissolved oxygen (DO ($mg\,L^{-1}$)) at an interval of 15 minutes. In-situ pH was measured at each site with an ion-sensitive field effect transistor sensor (Seabird SeaFET) at an interval of 5 minutes, with

an initial accuracy of ± 0.05 pH on the Total scale. In order to assess the sensitivity of NEP and NEC to light availability, we recorded photosynthetically active radiation at the seagrass canopy (PAR; µEinstein $m^{-2}\,s^{-1}$ [$\mu E\,m^{-2}\,s^{-1}$]) with a submerged Seabird ECO-PAR sensor equipped with an automatic wiper for the optics. We also deployed Lowell tilt current meters

(TCMs) at both of our primary sites to assess lateral transfer of water through the site, but the observed current speeds were below the minimum detectable speed for these instruments ($< \sim 2$ cm s$^{-1}$).

At the four FCE-LTER sites (Fig. 1), we measured DO and pH over a span of 4-7 days in September (BA, LM, and DK) and 8 days in December (SB), with an hourly sampling frequency using YSI EXO-2 sondes. These sites span broad gradients in phosphorus-limitation, seagrass productivity (Fourqurean et al. 1992), carbonate production (Yates and Halley 2006), DIC and TA concentrations (Millero et al., 2001), air-water $CO_2$ exchange (Yates and Halley 2006; DuFore 2012). We used this pH and DO data to calculate temporal excursions in DO ($\Delta$DO) and hydrogen ion concentration ($\Delta[H^+]$) (mM hr$^{-1}$), which are proxies for NEP$_{DO}$ and NEP$_{DIC}$ respectively (Long et al., 2015b). Data from these FCE-LTER deployments was compared with data from the two primary sites to determine whether the results of this study were generalizable to the rest of Florida Bay.

## 2.5 Benthic Chamber Fluxes

During the second sampling campaign, benthic chambers were deployed continuously over bare sediment at each of our primary sites to measure sediment-water fluxes of TA and DIC, excluding the effect of seagrass shoots. At the beginning of the experiment, acrylic chambers ($\sim$2.5L) were flushed with site water and placed at a naturally seagrass-free location on the sediment, within a few meters of each of our primary sites. Chamber incubations ran for a total of 4 days. At intervals ranging from 8-20 hr, $\sim$150 mL samples were taken from the chambers using a syringe, and the chambers were re-equilibrated with ambient site water. Fluxes were calculated based on the difference in concentration between the ambient water sample at the initial time of chamber placement, and the final concentration inside the chamber.

## 2.6 Sample Analysis

TA was analysed in at least triplicate (n = 3 to 5) 25 mL subsamples by automated Gran titration at a controlled temperature on an Apollo AS-ALK2, with an average precision (standard deviation of replicate measurements) of $\pm$1.89 $\mu$mol kg$^{-1}$ or 0.07% of the average measured TA. Samples for DIC were analysed by injecting 250 $\mu$L subsamples into an impinger filled with 10% HCl, converting all DIC to $CO_2$, which was subsequently transferred with a pure $N_2$ carrier gas to a LI-COR 6262 infrared gas analyser in integration mode. Samples were repeated injected (3-5 times) to improve the precision, which was still noticeably lower than that for TA, at $\pm$ 5.11 $\mu$mol kg$^{-1}$ or 0.21%. During each TA and DIC run, a certified reference material (CRM) was repeatedly measured to quantify any drift or systematic bias with these analyses. The CRM used was purchased from Dr Andrew Dickson at the Marine Physical Laboratory in La Jolla, California, and was a part of batch #154. We used these CRM measurements to correct TA and DIC, assuming a linear drift between repeat CRM runs. The magnitude of this correction was on average 0.75% for DIC and 0.34% for TA. Both TA and DIC measurements were converted to gravimetric units by multiplying the concentration ($\mu$M) by the calculated SSS and SST-derived seawater density using the Gibbs Seawater toolbox for Matlab (GSW; McDougall and Barker 2011) to derive units of $\mu$mol kg$^{-1}$.

Samples for $\delta^{13}C_{DIC}$ were analysed on a Thermo Gas Bench coupled to a Thermo Delta V Isotope Ratio Mass Spectrometer and reported in delta ($\delta$) notation in units of per-mille (‰) relative to Vienna Pee Dee Belemnite. Precision for this measurement was ±0.4‰ based on replicate analyses of Certified Reference Material (Dickson et al. 2003).

## 2.7 NEP and NEC Calculations

NEC, $NEP_{DIC}$, and $NEP_{DO}$ were determined by integrating temporal excursions in salinity-normalized TA ($n$TA), DIC ($n$DIC), and DO. We quantified the total TA or DIC inventory over time to determine NEC and NEP, in what is an application of the 'open water' approach. This approach requires a static water mass that is thoroughly mixed, and a water residence time that is sufficiently long to prevent lateral exchanges from affecting TA and DIC concentrations. This open water approach is often applied to shallow coastal systems including tidally-inundated coral reef lagoons which are restricted from exchanges with the coastal ocean at low tide (Shaw et al., 2012; McMahon et al., 2018). While this approach may not be appropriate for coral reef lagoons at high tide due to excessive lateral mixing and vertical heterogeneities (McMahon et al., 2018), this region in Florida Bay is not subject to tidally-driven mixing to the same extent. First, NEC (mmol $CaCO_3$ $m^{-2}$ $hr^{-1}$) was estimated using the alkalinity anomaly technique, which assumes that variations in TA are affected only by $CaCO_3$ precipitation and dissolution (1):

$$NEC = -0.5 \times \frac{\Delta n\text{TA}}{\Delta t} \times h\rho, \qquad (1)$$

where $\Delta n$TA was the difference in $n$TA ($n$TA = TA$\times$SSS$_{Average}$/SSS), $h$ the water depth, and $\rho$ the seawater density. The -0.5 scalar was required because 2 moles of TA are required to form one mole of $CaCO_3$ production. Salinity normalized DIC ($\Delta n$DIC) was calculated in the same manner as $\Delta n$TA. The temporal excursion in $n$TA used for Eq. 1 was calculated between each sampling point shown in Fig. 2g and 2h, for a total of 28 individual measurements of NEC. SSS$_{Average}$ was determined for each sampling campaign at each site. By convention, NEC is positive when TA consumption occurs and $CaCO_3$ is inferred to have been precipitated. Because of this, other processes which act as sources or sinks of TA will necessarily impact calculated NEC. Such processes include denitrification, which is a net source of TA due to the consumption of $HNO_3$. Sulfate reduction also produces TA, but only if reduced sulfur is retained in the sediment and is not oxidized in oxygenated pore-water. $NEP_{DO}$ (eq 2; mmol $O_2$ $m^{-2}$ $hr^{-1}$) and $NEP_{DIC}$ (eq 3; mmol C $m^{-2}$ $hr^{-1}$) were calculated in a similar manner, but with additional corrections for air-water gas exchange and DIC consumption by NEC:

$$NEP_{DO} = \frac{\Delta DO}{\Delta t} h\rho - O_2 \text{ Flux}, \qquad (2)$$

$$NEP_{DIC} = \frac{\Delta n\text{DIC}}{\Delta t} h\rho - NEC - CO_2 \text{ Flux}, \qquad (3)$$

where $O_2$ and $CO_2$ fluxes (eq 4 and 5) were estimated with a bulk-transfer approach using two different formulations for the gas transfer velocity ($k_{600}$; cm $hr^{-1}$). These $k_{600}$ parameterizations were intended to represent upper (Raymond and Cole (2001))

and lower (Ho et al., 2006) bounds for gas exchange, respectively. Wind data used to derive the $k_{600}$ were taken from the NOAA meteorological station at Islamorada (DW1872; Fig 1) and normalized to a height of 10m above the sea surface under neutral drag conditions ($U_{10}$; Large and Pond 1981).

$$O_2 \text{ Flux} = k_{600} * Sc * (O_2(water) - O_2(air)), \tag{4}$$

$$5 \quad CO_2 \text{ Flux} = k_{600} * Sc * K * (pCO_2(water) - pCO_2(air)), \tag{5}$$

where $pCO_{2(water)}$ was the partial pressure of $CO_2$ (µatm), and $O_2$ was the measured DO concentration (mg L$^{-1}$). $pCO_{2(water)}$ was calculated from measured TA and DIC using CO2SYS as above. Atmospheric $pCO_2$ ($pCO_{2(air)}$) was taken from the nearby Cheeca Rocks Mooring buoy operated by NOAA (Fig 1), while $O_{2(air)}$ was calculated from the measured DO (%). The gas solubility (K) and Schmidt numbers (Sc) were calculated from in-situ SSS and SST (Wanninkhof 1992; Weiss 1974). No

attempt was made to refine NEC by accounting for the TA produced by ecosystem productivity, but preliminary calculations assuming TA increases with DIC consumption at a ratio of 17/106 (Middelburg 2019) indicated that this TA production was a small fraction of total NEC (average difference of <10%). Furthermore, the implicit consideration of NEP$_{DIC}$ into the calculation of NEC (Eq. 1) introduces a circular reference in Eq. 3 (which includes NEC) that cannot be resolved in this approach.

**2.8 Uncertainty analysis for NEP and NEC calculations**

While our primary study sites are minimally affected by lunar tides, light water currents driven by wind and other factors do occur. When current speed is sufficiently high, and combined with spatial gradients in TA or DIC, the assumptions implicit in the 'open water' approach may not hold, and calculated metabolic rates will be subject to error. We consider this advection combined with spatial concentration gradients to be the largest source of uncertainty in our metabolic calculations.

To address this concern, we calculated upper and lower bounds of NEC and NEP using conservative estimates of possible advective TA, DIC, and DO exchange. Given the spatial separation between the high- and low-density sites of approximately 4 km, and the average concentration difference in TA of 300 µmol kg$^{-1}$, we estimate an average spatial gradient of 300/4, or 75 µmol kg$^{-1}$ km$^{-1}$. Given the close relationship between TA and DIC at this site, we consider the spatial gradient in DIC equal to that for TA. The average spatial gradient in DO was much lower, at 4.6 µmol kg$^{-1}$ km$^{-1}$. These spatial concentration gradients

($\frac{\Delta TA}{\Delta x}, \frac{\Delta DIC}{\Delta x}, \frac{\Delta DO}{\Delta x}$) were combined with a conservative estimate of water velocity ($u$) of 1.0 cm s$^{-1}$ to estimate the contribution of advective forcing to calculated metabolic rates. Because current speed was below the limit of detection, we cannot infer current direction, leading us to take the cautious approach of applying this error term as an absolute value to both sides of our metabolic rate measurements. For example, the upper (NEC$_{UB}$) and lower bounds for NEC (NEC$_{LB}$) were calculated as: $NEC_{LB} = -0.5 \times (\frac{\Delta nTA}{\Delta t} - [u \times \frac{\Delta TA}{\Delta x}]) \times h\rho$, and $NEC_{UB} = -0.5 \times (\frac{\Delta nTA}{\Delta t} + [u \times \frac{\Delta TA}{\Delta x}]) \times h\rho$. Uncertainty bounds for NEP$_{DIC}$ and NEP$_{DO}$

were calculated in the same manner, using average spatial gradients in DIC and DO listed above.

## 3. Results

## 3. Physico-chemical conditions

At each site, variations in SSS were generally less than 1 during each sampling campaign, indicating that precipitation and fresh groundwater inputs were likely minor sources of fresh water to these sites during the study period (Fig 2c,d). Across sampling campaigns, SSS was more variable, ranging from 33.15 to 34.63 at the high-density site, and from 31.45 to 34.67 at the low-density site. SST at both sites tracked each other closely, exhibiting diurnal variations of ~2 °C, and ranging from 18.5 to 27.0 across the entire study period (Fig. 2c,d). Diurnal variations in PAR coincided with those in SST, as is typical for sun-lit shallow water (Fig 2k,l). Likewise, both DO and pH exhibited typical diel excursions. Peak DO concentration of 8.14 (High-density) and 9.45 mg $L^{-1}$ (Low-density) occurred in the late afternoon, coinciding with maximum pH of approximately 8.17 (High-density) and 8.29 (Low-density) respectively. Average pH was $8.08 \pm 0.05$ at the high-density site, compared with $8.17 \pm 0.05$ at the low-density site. Calculated $pCO_{2(water)}$ at the high-density site ($538.8 \pm 123.5$ µatm) was generally greater than atmospheric equilibrium, while average $pCO_{2(water)}$ was less than $pCO_{2(air)}$ at the low-density site ($390.3 \pm 129.4$) (Table 1). Calculated $CO_2$ flux was generally positive (from the water to the atmosphere) and small in magnitude, between $0.13 \pm 0.62$ and $0.38 \pm 0.20$ mmol C $m^{-2}$ $hr^{-1}$ at the high-density site (RC01 and Ho06 respectively), and $0.20 \pm 0.40$ and $0.067 \pm 0.35$ mmol C $m^{-2}$ $hr^{-1}$ at the low-density site (Table 1). There was a difference between $CO_2$ fluxes derived using the RC01 and Ho06 $k_{600}$ parameterizations, but this difference was small in magnitude compared to NEP and NEC, so for the sake of simplicity, we only present results using the Ho06 parameterization in the main text of this manuscript. Results considering both parameterizations are given in the supporting information.

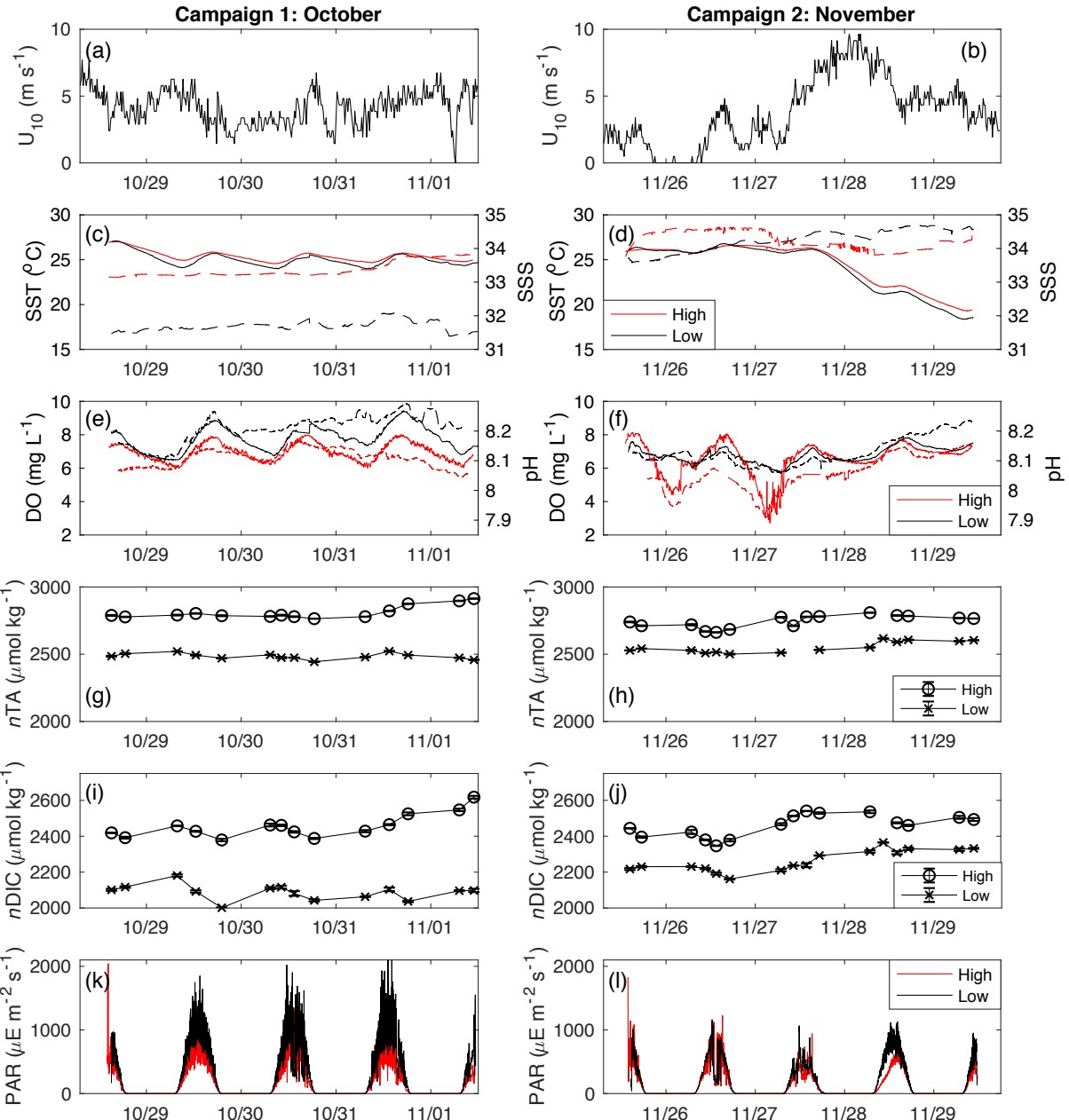

**Figure 2.** Time-series of a-b) $U_{10}$ (m s$^{-1}$), c-d) SST and SSS, e-f) DO (mg L$^{-1}$) and SeaFET pH, g-h) $n$TA (µmol kg$^{-1}$), i-j) $n$DIC (µmol kg$^{-1}$), and k-l) PAR (µE m$^{-2}$ s$^{-1}$). For plots c-f, the solid lines are linked to the left axis, while the dashed lines are for the right axis.

**Table 1.** Table of physicochemical conditions (TA, DIC, Salinity), as well as seagrass and sediment chemical characteristics (average ± SD).

| | High Density | Low Density |
|---|---|---|
| Aerial Productivity (g m$^{-2}$ d$^{-1}$) | 0.59 ± 0.26 (n=6) | 0.41 ± 0.36 (n=6) |
| Sediment C$_{org}$ (% of dry weight) | 5.8 | 1.4 |
| Sediment C$_{inorg}$ (% of dry weight) | 7.9 | 10.1 |
| C$_{org}$ : C$_{inorg}$ | 0.74 | 0.14 |
| Sediment Depth (cm) | 56 ± 15 (n 10) | 32 ± 5 (n 10) |
| Above-ground Biomass (g m$^{-2}$) | 65.11 ± 17.66 (n=6) | 15.09 ± 14.46 (n=6) |
| Salinity | 33.8 ± 0.49 | 33.0 ± 1.3 |
| DIC (µmol kg$^{-1}$) | 2489.4 ± 74.7 | 2212.2 ± 134.0 |
| TA (µmol kg$^{-1}$) | 2810.6 ± 51.4 | 2550.5 ± 83.2 |
| Water Depth (m) | 2.1 | 1.7 |

| | High Density | Low Density |
|---|---|---|
| $\Omega_{calcite}$ | 5.83 ± 0.84 | 6.23 ± 1.15 |
| $p$CO$_2$ (µatm) | 538.8 ± 123.5 | 390.3 ± 129.4 |
| pH | 8.10 ± 0.055 | 8.17 ± 0.062 |
| CO$_2$ Flux–Ho06 (mmol m$^{-2}$ hr$^{-1}$) | 0.38 ± 0.20 | 0.13 ± 0.62 |
| O$_2$ Flux–Ho06 (mmol m$^{-2}$ hr$^{-1}$) | 0.034 ± 1.2 | 0.75 ± 1.9 |
| Seagrass N:P (mol:mol) | 82.7 | 102.1 |
| Seagrass C:P (mol:mol) | 1303.4 | 1892.7 |
| Seagrass C:N (mol:mol) | 15.8 | 18.5 |
| Sediment N:P (mol:mol) | 12.3 | 8.3 |
| Sediment C:P (mol:mol) | 321.8 | 1187.4 |
| Sediment C:N (mol:mol) | 26.3 | 142.3 |

Between the first and second sampling campaigns, average mid-day PAR (from 10:00 to 14:00) reaching the benthos at the low-density site fell by approximately 38%, from 916 ± 332 m$^{-2}$ s$^{-1}$ during the first sampling campaign to 567 ± 219 µE m$^{-2}$ s$^{-1}$ for the second sampling campaign. Similarly, average mid-day PAR at the high-density site fell by ~31%, from 627 ± 259 µE m$^{-2}$ s$^{-1}$ during the first sampling campaign, to 432 ± 211 µE m$^{-2}$ s$^{-1}$ for the second sampling campaign. After the passage of a large cold front and associated high wind speed on 11/28, SST fell by more than 5 °C. At the initial SSS, DIC, and TA, the thermodynamic effect of this cooling was a nearly 0.1 increase in pH (CO2Sys), which was on the order of the typical diel range (Fig 2e,f). While this rapid pH increase (independent of DO) was evident at the low-density site, no such change occurred at the high-density site (Fig 2f), indicating that biological factors outweighed the thermodynamic effect on pH there.

Across the study period, $n$TA at the high-density site was always greater than $n$TA at the low-density site, and $n$TA was generally higher than $n$DIC at both sites. Diel cycles were evident in both $n$DIC and $n$TA, coinciding with typical variations in net ecosystem production (consuming $n$DIC), and calcification (consuming $n$TA). The average slope between $n$TA and $n$DIC ($\Delta n$TA:$\Delta n$DIC) was 0.64 and 0.41 for high- and low-density sites respectively (Fig 3), indicating that variations in TA and DIC were likely driven by a combination of ecosystem metabolism (expected slope of -0.15 if $NO_3$ is used), calcification (slope of 2), as well as $SO_4^{2-}$ reduction (slope of 1) and denitrification (slope of 0.8), as has been suggested for other Florida seagrasses (Camp et al., 2016; Challener et al., 2016). However, in this underdetermined case in which all of the aforementioned processes are occurring, the application of a simple $n$TA vs $n$DIC plot cannot reveal the relative importance of these factors.

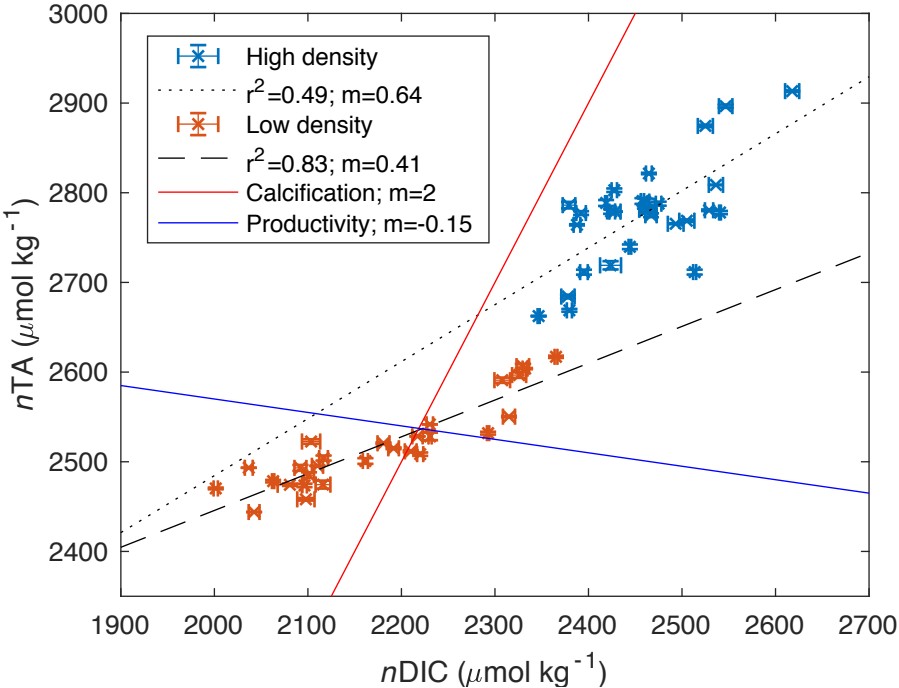

**Figure 3.** Scatter plot of $n$DIC and $n$TA for both high-density (blue) and low-density (orange) sites, and associated slope (m) and correlation coefficient ($R^2$) of the linear regression. The red reference line indicates the expected relationship if calcification is dominant, consuming 2 moles of TA for every mole of DIC consumed to form $CaCO_3$. The blue reference line shows the approximate relationship expected for aerobic respiration/productivity, which consumes approximately 0.15 moles of TA for every mole of DIC respired.

**3.2 NEP and NEC**

At both sites, calculated $NEP_{DO}$ and $NEP_{DIC}$ followed a clear diel pattern, increasing between sunrise and early afternoon, and decreasing through sunset (Fig. 4). Night-time $NEP_{DO}$ and $NEP_{DIC}$ was nearly always negative (heterotrophic),

while daytime values were larger and more variable, often exceeding ~15-20 mmol C $m^{-2}$ $h^{-1}$ in the late morning. Individual measurements of $NEP_{DIC}$ for the low- density site (-14.5 to 29.2 mmol C $m^{-2}$ $h^{-1}$) and high-density site (-36.2 to 21.4 mmol C $m^{-2}$ $h^{-1}$) were very large compared with seagrass aboveground primary productivity, which was between 1.5-2 μmol C $m^{-2}$ $h^{-1}$ at both sites (Table 1). While NEC was also strongly negative (dissolving) at night, it was highly variable during the day, with

5  no clear trend between sunrise and sunset (Fig 4). It is important to note that this approach does not account for any TA production by net $SO_4^2$ reduction and denitrification, and any such TA inputs may bias these estimates of NEC. However, our NEC estimates are at least an order of magnitude larger than typical published measurements of seagrass $SO_4^{2-}$ reduction (Holmer et al., 2003; Brodersen et al., 2019) and denitrification (Welsh et al., 2001) rates, suggesting that our NEC determinations were indeed largely driven by $CaCO_3$ precipitation and dissolution. Still, other studies have found relatively

10  high rates of $SO_4^{2-}$ reduction in seagrass sediments (Hines and Lyons 2007), especially those with high seagrass shoot density (Holmer and Nielsen, 1997), so we express caution in the interpretation of our NEC results.

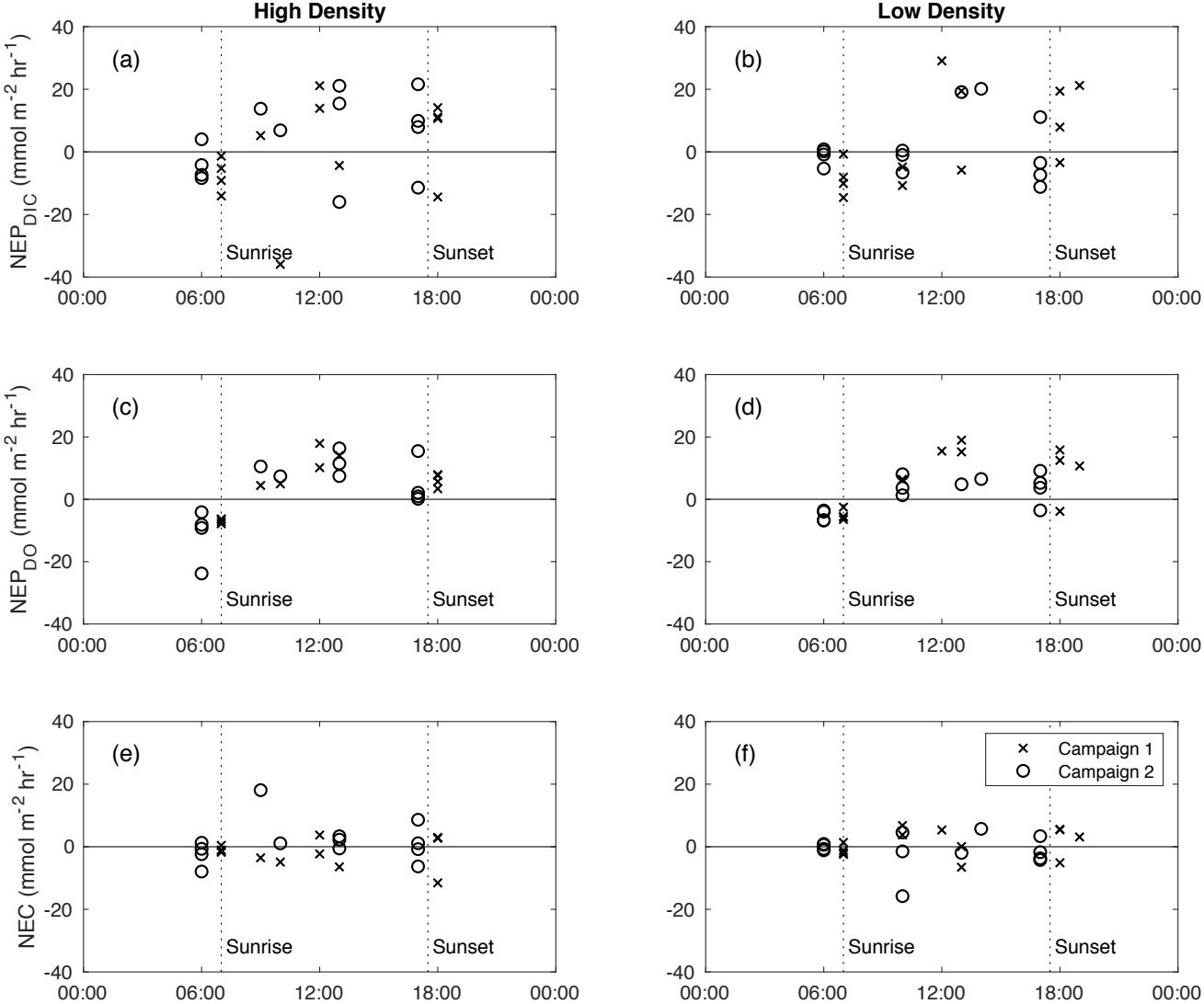

**Figure 4.** Diel trends in NEP$_{DIC}$ (a-b) and NEP$_{DO}$ (c-d) NEC (e-f), for the high-density site (left panels) and low-density site (right panels), for sampling campaign 1(X) and 2 (O). The x-axis represents the midpoint time for each NEP or NEC calculation period.

    As discussed previously, the combination of advection with spatial concentration gradients can generate an error in calculated metabolic rates by breaking the assumptions required in the 'open water' approach. When NEP$_{DIC}$ or NEC were large, our estimated uncertainty due to this mixing effect was relatively low (Fig 5). However, when metabolic rates were close to zero, the effect of advection became quite large and potentially problematic. The average uncertainty in NEC due to

10   advection ($u \times \frac{\Delta TA}{\Delta x}$) was 2.4 and 2.9 mmol CaCO$_3$ m$^{-2}$ h$^{-1}$ for the low- and high-density sites respectively. This corresponds

to 65 and 76 % of average NEC. Likewise, this mixing error could account for 4.7 and 5.8 mmol C m$^{-2}$ h$^{-1}$, or ~50 % of average NEP$_{DIC}$. While this effect was at times large for both NEP$_{DIC}$ and NEC, it was quite small for NEP$_{DO}$, due to the small spatial gradients in DO present between our two primary sites. The uncertainty in NEP$_{DO}$ due to advection was 0.28 and 0.34 mmol O$_2$ m$^{-2}$ h$^{-1}$, or 4.0 and 4.2 % of average rates at the low- and high-density sites respectively.

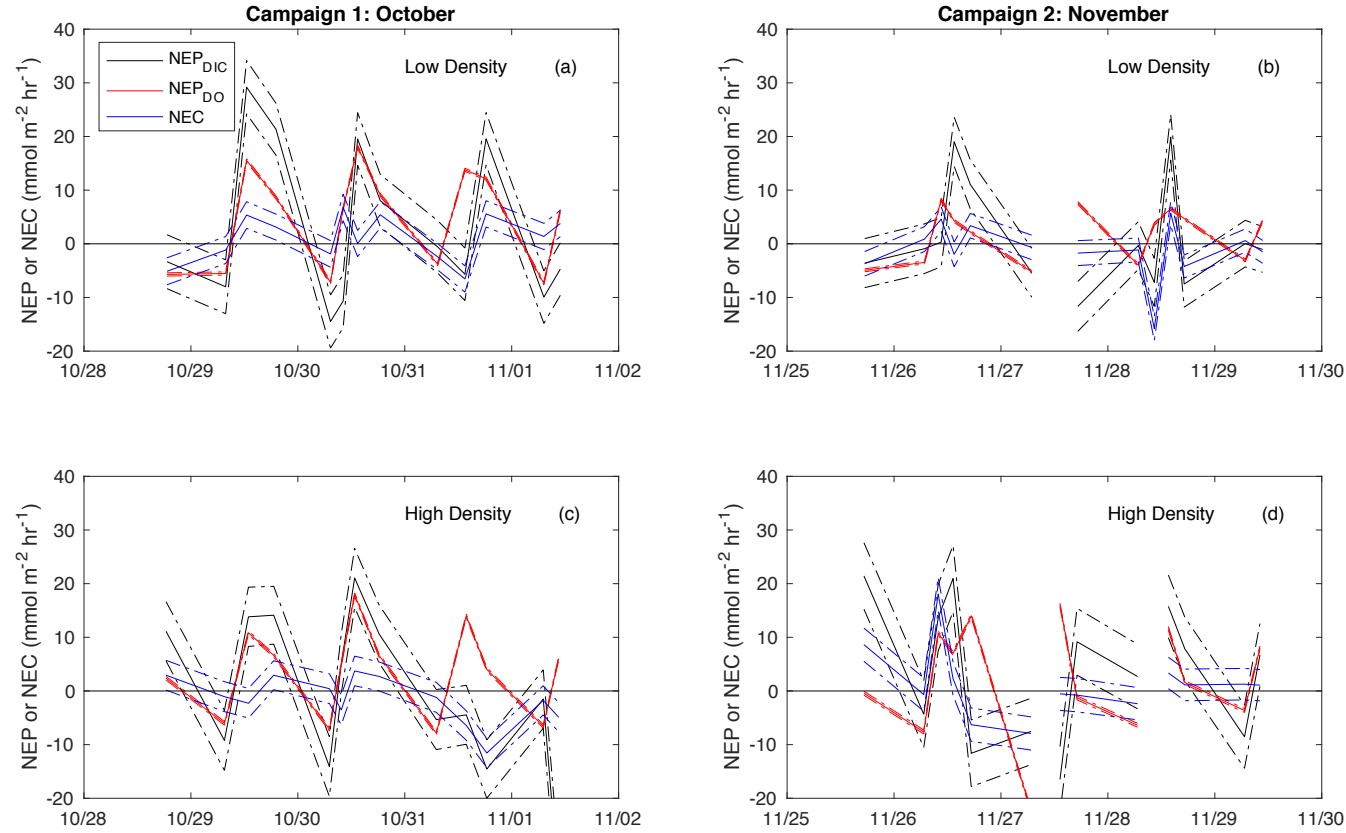

**Figure 5.** Time-series plot of NEC (blue), NEP$_{DIC}$ (black) and NEP$_{DO}$ (red), including the upper and lower uncertainty bounds related to error due to advection (dashed lines).

When discrete NEP and NEC rates were integrated over cumulative day and night hours, diel trends became more recognizable (Fig. 6a-b). NEP$_{DIC}$ and NEP$_{DO}$ was positive during the day (net autotrophic) and negative (net heterotrophic) at night for both sites. While the impact of advective exchanges on the uncertainty of metabolic calculations was minor for NEP$_{DO}$, it was relatively important for NEP$_{DIC}$. While mean daytime NEP$_{DIC}$ was positive at both sites, the estimated lower bounds for daytime NEP$_{DIC}$ were slightly negative, at -0.03 and -1.1 mmol C m$^{-2}$ h$^{-1}$ for the low- and high-density sites respectively. This is partially due to the act of binning metabolic values by 'day' and 'night', which combines early morning and afternoon rates with mid-day peaks in NEP and NEC. Metabolic rates did not (and should not) exhibit a step-wise change during sunrise and sunset, but data from these time periods was combined with mid-day peaks in NEP and NEC in Fig 4. In

other words, temporally integrating by day and night over a sinusoidal diel signal will always have the effect of decreasing the absolute magnitude of average metabolic rates for day and night time periods. However, we emphasize that this simple uncertainty analysis gives us ample reason to be cautious when interpreting metabolic rates derived from 'open water' approaches in coastal waters. Nevertheless, both mean night-time $NEP_{DIC}$, as well as its upper and lower bounds were negative,

giving strong evidence that these sites were indeed net heterotrophic at night (Fig 6a), and net heterotrophic over the study period (Fig 6b).

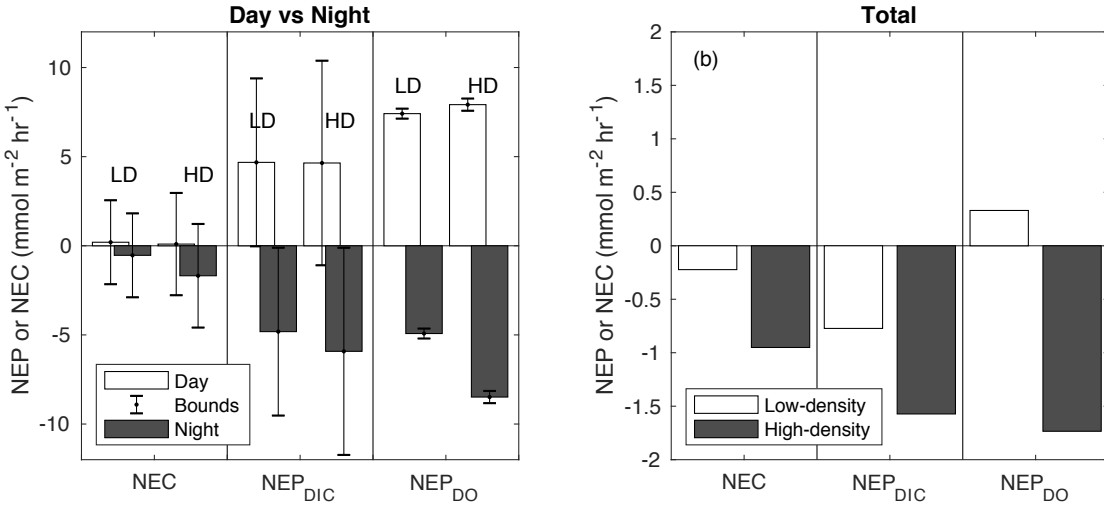

**Figure 6.** NEC, $NEP_{DIC}$, and $NEP_{DO}$ integrated over day and night-time periods (a), and over the entire study period (b). NEP values are shown for $k_{600}$ of Ho et al., 2006. The error bars in figure (a) represent upper and lower bounds for metabolic rates

determined in section 2.8.

While the advective uncertainty term for NEC calculations was similar in size to that for $NEP_{DIC}$, rates of NEC were typically lower than $NEP_{DIC}$, causing upper and lower uncertainty bounds for day and night integrated NEC to contain zero (Fig 6a). This was due to the obscuring effect of integrating over day and night periods, as well as the choice of a highly conservative estimate of water velocity (1 cm s$^{-1}$) in this uncertainty analysis. The presence of larger spatial concentration

gradients, faster currents, or greater water depth could all cause this uncertainty term to increase in relation to metabolic rates. Nevertheless, in this study, NEC was more consistently negative (net dissolving) at night (Fig. 6a), causing cumulative NEC to be less than zero (Fig. 6b). This night-time dissolution was slightly greater at the high-density site than the low-density site. Given the relative paucity of positive NEC estimates across the study period (Fig 4) and the clear signal of negative NEC during the night, it is likely that net dissolving conditions were more frequent than net calcifying conditions. Therefore, we

have confidence that over the full study period, both sites were net dissolving (-NEC), as depicted in Fig 6b. Average NEC was less than $NEP_{DIC}$, such that the NEC:$NEP_{DIC}$ ratio was 0.54 and 0.31 for the high- and low-density sites respectively, well within the range of tropical seagrass ecosystems globally (Camp et al., 2016) and locally (Turk et al., 2015).

While NEP$_{DIC}$ and NEC were likely negative (heterotrophic and dissolving) at both sites over the entire study period (Fig 6b), NEP$_{DO}$ was small and positive at the low-density site, and negative at the high-density site. This difference between NEP$_{DO}$ and NEP$_{DIC}$ was still prominent when values were split by day and night. Although NEP$_{DIC}$ and NEP$_{DO}$ agreed in direction, NEP$_{DO}$ was greater in magnitude than NEP$_{DIC}$ for all time periods except at night for the low-density site (Fig 5a). In fact, the linear relationship between NEP$_{DO}$ and NEP$_{DIC}$ in this study was not significantly different from 0 for the high-density site (p=0.095; r$^2$=0.11) and was significant but weak (p=0.001; R$^2$=0.35) for the low-density seagrass site (Fig 8). While NEP$_{DO}$ and NEP$_{DIC}$ agreed in sign at night (dark blue points in Fig 8), there was no such relationship for daytime NEP$_{DO}$ and NEP$_{DIC}$. Correlations between net ecosystem processes and PAR were not strong (R$^2$<0.5) for NEP$_{DIC}$ and NEP$_{DO}$ and were very weak (R$^2$<0.05) for NEC (Fig. 7a-c).

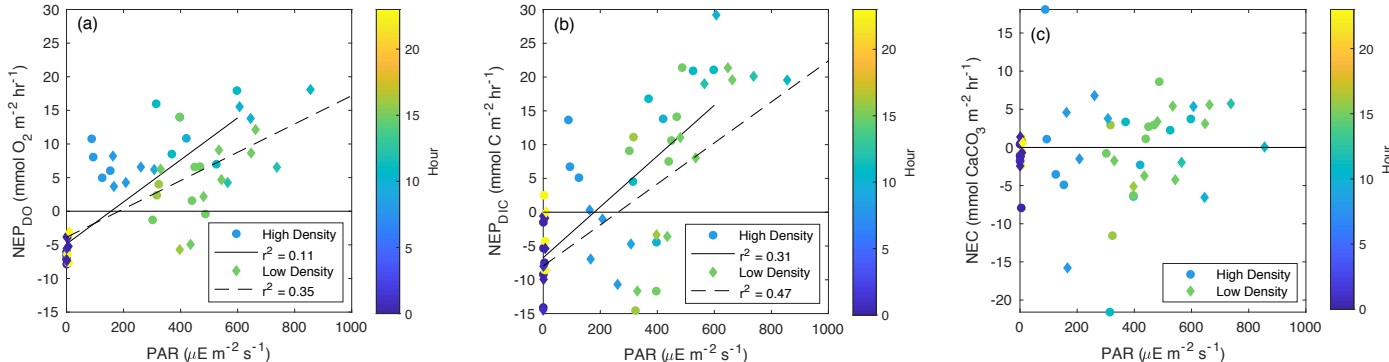

**Figure 7.** Scatter plots of (a) NEP$_{DO}$ vs PAR, NEP$_{DIC}$, and NEC vs PAR (b-c). Points are colored by the average hour for the respective time period over which NEP or NEC was calculated. The arrows in (a) are intended to highlight the hysteretic pattern between PAR and NEP$_{DO}$.

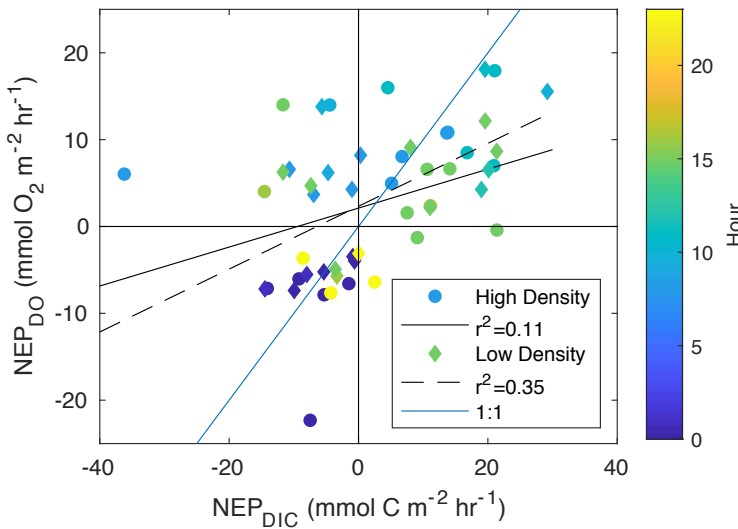

**Figure 8.** Scatter plots of $NEP_{DO}$ vs $NEP_{DIC}$.

To address whether this disconnect between $NEP_{DO}$ and $NEP_{DIC}$ exists outside of the two primary sites (Fig. 9; High- and Low-Density sites), we assembled pH and DO data from 4 additional sites across Florida Bay (Fig. 9: SB, BA, DK, and LM). Even though $\Delta[H^+]$ and $\Delta DO$ were correlated at our primary sites and one of the four LTER sites (LM), correlations were poor ($R^2 < 0.25$) at the remaining LTER sites. The LM site is heavily influenced by terrestrial inputs from the coastal Everglades and fringing mangroves, which likely contributed to the significant relationship between $\Delta[H^+]$ and $\Delta DO$ there ($R^2 = 0.48$).

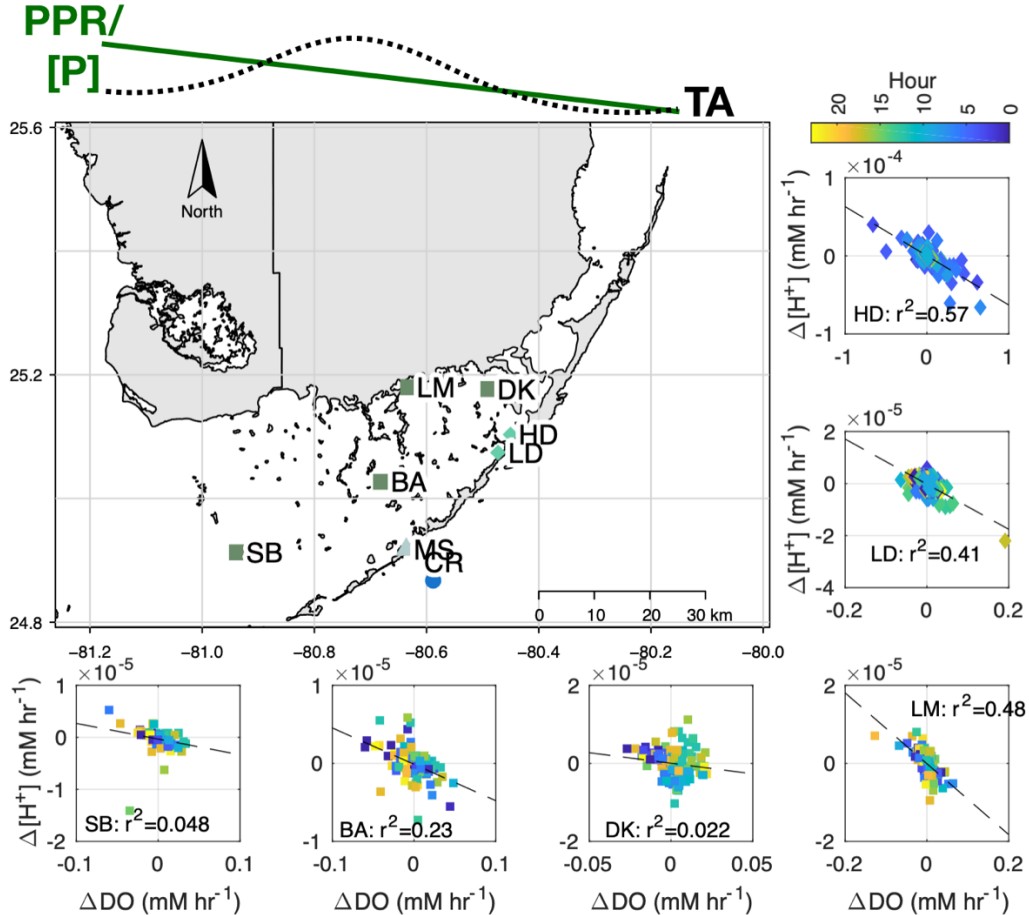

**Figure 9.** Map showing $\Delta[H^+]$ vs $\Delta DO$ relationship for sites associated with LTER (SB, BA, DK, LM) and the present study (high-density [HD] and low-density [LD]). At the top of the figure, we present the general east-to-west pattern in seagrass primary productivity (PPR), phosphorus content ([P]; Fourqurean et al., 1992), and TA (Millero et al., 2001) within Florida Bay. All LTER sites failed to meet the assumptions for a test of slope significance (gvlma package in R), so we simply report the $R^2$.

### 3.3 $\delta^{13}C_{DIC}$ and benthic flux of TA and DIC

While both sites were net dissolving (-NEC) over the study period (Fig. 6b), the calculated calcite saturation state ($\Omega_{calcite}$, CO2Sys) was relatively high, at $5.83 \pm 0.84$ and $6.23 \pm 1.15$ at the high- and low-density sites, respectively (Table 1), indicating that dissolution of carbonates in the sediments was contributing to water column DIC. The uncertainty of this $\Omega_{calcite}$ calculation was $\pm 0.30$, or approximately 5% of the average value. The 'Keeling plot' indicated source $\delta^{13}C_{DIC}$ values were -$6.9 \pm 3.7$ and $-8.8 \pm 6.8$ ‰ (95% confidence interval) for the high- and low-density sites respectively (Fig. 10).

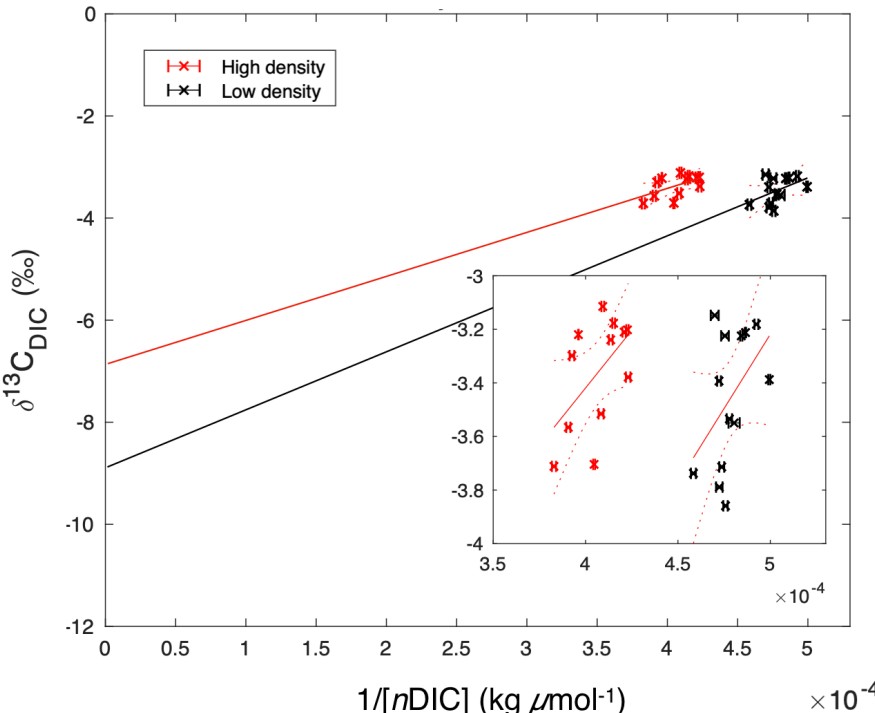

**Figure 10.** 'Keeling plot' of 1/[$n$DIC] versus $\delta^{13}C_{DIC}$, suggesting potential end-member isotopic values. These y-intercept $\delta^{13}C_{DIC}$ values were -6.9 ± 3.7 and -8.8 ± 6.8 ‰ (95% confidence interval) for the high- and low-density sites respectively. The inset figure is zoomed to the extent of collected data, while the large figure is scaled to demonstrate the extrapolation required in order to extend the data to the y-intercept.

Benthic chamber flux experiments (over bare sediment) during the second sampling campaign yielded average benthic DIC fluxes of 0.76 ± 0.7 and 1.26 ± 0.8 mmol m$^{-2}$ h$^{-1}$ at the low- and high-density sites, respectively. These benthic DIC fluxes could explain 109% (0.76/-0.7 = 1.09) of the average NEP$_{DIC}$ at the low-density site, and 79% (1.26/-1.6 = 0.79) at the high-density site. Benthic TA fluxes were 0.24 ± 0.16 mmol m$^{-2}$ h$^{-1}$ at the low-density site but were highly variable and not significantly different from zero at the high-density site (0.16 ± 0.4 mmol m$^{-2}$ h$^{-1}$). Benthic TA flux could explain 120% (0.24/-0.2 = 1.2) of cumulative NEC at the low-density site, but only 18% (0.16/-0.9 = 0.18) at the high-density site.

## 4. Discussion

### 4.1 Drivers of NEP

Individual measurements of NEP$_{DIC}$ for the low- density site (-14.5 to 29.2 mmol C m$^{-2}$ h$^{-1}$) and high-density site (-36.2 to 21.4 mmol C m$^{-2}$ h$^{-1}$) were within the range of some previous studies, including NEP$_{DO}$ from of Turk et al. 2015 (−6.2 ± 1.0 to 12.3 ± 1.0 mmol O$_2$ m$^{-2}$ h$^{-1}$), Perez et al. 2018 (~ 23.8 mmol O$_2$ m$^{-2}$ h$^{-1}$) and Long et al. 2015a (0.45-1.46 mmol O$_2$

m$^{-2}$ h$^{-1}$). Over the entire study period, however, cumulative NEP$_{DIC}$ was negative at both sites (Fig. 6b), indicating that heterotrophic conditions dominated in both seagrass meadows during these two sampling campaigns. CO$_2$ fluxes were positive at both sites, indicating a net release of CO$_2$ from the water to the atmosphere (Table 1). Seagrass aboveground primary productivity rates were between 1.5-2 μmol C m$^{-2}$ h$^{-1}$ at both sites (Table 1), approximately 3 orders of magnitude lower, and opposite in sign, than the measured NEP$_{DIC}$. This large difference provides further evidence that seagrass aboveground primary productivity is only a component of net ecosystem productivity, which was likely dominated by sediment processes (including seagrass belowground productivity, which was not measured during this study), and epiphytic primary productivity. We found a clear disagreement between daytime NEP$_{DO}$ and NEP$_{DIC}$, such that the linear relationship between NEP$_{DO}$ and NEP$_{DIC}$ was not significantly different from 0 for the high-density site (p=0.095; R$^2$=0.11) and was significant but very weak (p=0.001; R$^2$=0.35) for the low-density site (Fig 8). Such a disagreement between NEP$_{DO}$ and NEP$_{DIC}$ has been observed recently in coral ecosystems (Perez et al., 2018). This discrepancy between NEP$_{DO}$ and NEP$_{DIC}$ may be related to the thermodynamics of CO$_2$ and O$_2$ dissolution, as the solubility of O$_2$ is much less than that of CO$_2$ (Weiss 1970; 1974). Any O$_2$ produced or consumed by NEP will rapidly exchange with the atmosphere, while most of the CO$_2$ generated by NEP will enter the carbonate buffering system and persist as HCO$_3^-$ or CO$_3^{2-}$ ions, rather than exchangeable CO$_2$. The standard deviation of O$_2$ fluxes was much larger than that of CO$_2$ fluxes, in part due to this effect. Furthermore, as the total pool of O$_2$ in the water column is far less than the total pool of CO$_2$ (i.e. DIC), the determination of NEP$_{DO}$ is more sensitive to the parameterization of gas transfer than is NEP$_{DIC}$. This is highlighted in Fig S1, where the difference between the two k$_{600}$ parameterizations is much larger for NEP$_{DO}$ than for NEP$_{DIC}$.

Further explanations for this discrepancy between NEP$_{DO}$ and NEP$_{DIC}$ can be related to differing responses of DO and DIC to variations in light availability. When PAR was plotted against NEP$_{DO}$, a clear pattern of hysteresis arose, with higher NEP$_{DO}$ values during the morning hours than the afternoon at the same PAR intensity (shown by the arrows in Fig. 7a). Such a hysteretic pattern indicates that the response of NEP$_{DO}$ to light is not uniform, and that photosynthetic efficiency may vary with factors such as nutrient availability, history of carbon acquisition (carbon concentrating mechanisms) or temperature. Such a hysteretic pattern has been observed between PAR and NEC, but not for NEP, for a coral reef (Cyronak et al., 2013). This has important implications for the modeling of carbon processing in seagrass meadows, which generally assume a time-invariant relationship between light and photosynthesis (Zimmerman et al., 2015; Koweek et al., 2018).

### 4.2 Drivers of NEC

We found no relationship between PAR and NEC at our study sites, indicating that light-driven calcification by photoautotrophs (algal epiphytes, calcifying macroalgae and seagrasses themselves) does not dominate NEC, or that carbonate dissolution driven by respiration in the sediments dominated NEC. However, it is possible that the use of carbon concentrating mechanisms could cause calcification by photoautotrophs to become decoupled from direct irradiance. While not listed in Table 1, we did observe a variety of bivalves and tube-building polychaetes that may have contributed to the high NEC at both sites. Furthermore, while Ω$_{calcite}$ was always greater than 1, NEC was negative on average over the study period, indicating

that the overall ecosystem was net dissolving. This co-occurrence of high $\Omega_{calcite}$ with overall net dissolving conditions (-NEC) can be reconciled by considering the seagrass ecosystem as a vertically de-coupled system, where positive NEC in the water column is more than balanced by carbonate dissolution in the sediments. Such a relationship has been observed or inferred in seagrasses elsewhere (Millero 2001; Burdige and Zimmerman 2002; Burdige et al., 2010).

Our 'Keeling plot' approach indicated potential end-member $\delta^{13}C_{DIC}$ values that lie between the $\delta^{13}C$ of seagrass organic matter (~ -8 to -10 [Fourqurean et al., 2015; Röhr et al., 2018]) and sediment inorganic carbon (~0 ‰ [Deines 1980]), indicating that both sediment organic matter respiration and carbonate dissolution were sources of DIC. It should be noted that this approach involves the extension of measurements to a theoretical $\delta^{13}C_{DIC}$ value at infinite DIC concentration, involving a substantial extrapolation (Fig. 10). Furthermore, this isotopic analysis implicitly assumes a closed system, which clearly is

not the case in Florida Bay.

        From these lines of evidence, we infer that OC remineralization in sediments, combined with carbonate dissolution contributed to the net upward DIC and TA fluxes from the sediments, which appear to have driven the observed negative NEP (heterotrophy) and NEC (dissolution), respectively. Such net heterotrophy must be fuelled by $C_{org}$ captured by the system, either from allochthonous sources or from autochthonous sources occurring at some time in the past. This study was conducted

at two relatively deep-water sites during autumn with relatively low light levels and short days, so it is quite possible that there could be a different net annual signal when the bright summer months are included, highlighting the need for annually-resolved measurements. However, the results of our benthic flux experiments support the isotopic evidence for the role of sediment OM remineralization in NEP and NEC at these sites. When expressed as aerial fluxes, sediment-water DIC exchange was 79 and 109% of average $NEP_{DIC}$ at the high- and low-density sites, respectively. Likewise, benthic TA flux was 18-120% of

cumulative NEC. Together, these benthic flux measurements, along with isotopic evidence, supports the role of sediment biogeochemical cycling in the overall carbon budget at these sites. Prior studies have shown high rates of denitrification (Eyre and Ferguson 2002) and $SO_4^{2}$ reduction (Hines and Lyons 2007; Holmer et al., 2001; Smith et al., 2004) in seagrass soils, so it seems quite possible that these processes contributed to much of the inferred net ecosystem heterotrophy here. The extent to which these anaerobic TA-generating processes also affect our NEC estimates is largely dependent on the fraction of reduced

species that are re-oxidized in oxygenated micro-zones within surface sediments. There is a clear need for more research exploring the linkages between sediment early diagenesis and water-column biogeochemistry over seagrasses. This is especially important, given the recent attention that seagrass systems have received as potential 'buffering' mechanisms for coastal ocean acidification (Manzello et al., 2012; Unsworth et al., 2012; Hendriks et al., 2014; Cyronak et al., 2018; Koweek et al., 2018; Pacella et al., 2018).

However, there is a geologic context for this observed negative NEC in the northeast region of Florida Bay. Florida Bay is geologically young, having formed during the retreat of the Holocene shoreline following the end of the last major glaciation approximately 4-5,000 years before present (Bosence et al., 1985). The sedimentary deposits that filled in this basin are dominated by calcareous mud formed by extensive *Thalassia* meadows, and their associated epibionts and macroalgae (Bosence et al., 1985), and these autochthonous sources are sufficient to explain the observed sediment distributions (Stockman

et al., 1967). Early work suggests that calcareous sediments in Florida Bay can be separated into distinct zones of calcareous sediment formation, migration, and destruction, the last of which extends across NE Florida Bay, where this study took place (Wanless and Tagett et al., 1989). A limited sediment supply of ~0.01 mm yr$^{-1}$ in this 'destructional' zone, compared to the rate of sea level rise, results in the presence of a thin veneer of sediment on the bottoms of the basins and narrow, erosional mud banks (Stockman et al., 1967). Our primary sites were in this "destructional zone", and our finding of negative NEC indicates that at these sites (during the fall season), the "destructional" nature of this part of the bay may be partly explained by net carbonate dissolution. It is important to note the limited spatial and temporal scope of this study, and we caution that our findings of net negative NEP and NEC are likely not applicable to Florida Bay as a whole, or even to these sites across seasons. Indeed, prior studies have shown substantial seasonal and spatial variability in carbonate chemistry (Millero et al., 2001; Zhang and Fischer 2014) and seagrass primary productivity (Fourqurean et al., 2005).

Lastly, it is clear that sediments below seagrasses in Florida Bay have been accumulating autochthonous organic carbon ($C_{org}$) and carbonate sediments for over 3,000 years (Fourqurean et al. 2012b), suggesting that the ecosystem is producing more organic matter than it is consuming, and is storing more carbonates than it is dissolving (Howard et al., 2018). To reconcile our finding of net negative NEP and NEC with the knowledge that this system is a net producer of $C_{org}$ and $CaCO_3$, we must infer that NEP and NEC are not homogeneous throughout Florida Bay or throughout the year.

## 4.3 Regional Implications and Future Outlook

Variations in TA and DIC exports affect the carbonate buffering of adjacent ecosystems, further complicating the relationship between NEP$_{DO}$ and NEP$_{DIC}$. In Fig. 9, we show that correlations between $\Delta[H^+]$ and $\Delta DO$ at the LTER sites were generally poor. This poor fit is partially caused by the existence of the carbonate buffering system, which dampens the magnitude of pH variability, in comparison with the unbuffered nature of DO. However, we maintain that lateral TA transport also affects the relationship between $\Delta[H^+]$ and $\Delta DO$, given the phosphorus-driven spatial gradient in seagrass primary production in Florida Bay (Zieman et al., 1989; Fourqurean et al., 1992), and the realization that ecosystem production is linked with increased calcification (Frankovich and Zieman 1994; Enríquez and Schubert 2014; Perez et al., 2018). In addition, the mangroves that lie upstream of Florida Bay export water high in DIC and TA, and low in DO to Florida Bay (Ho et al., 2017), so that areas immediately affected by this runoff (like LTER site LM) will have a larger range in $\Delta[H^+]$ and $\Delta DO$. Likewise, we can infer that the relationship between NEP$_{DO}$ and NEP$_{DIC}$ is also altered by spatio-temporal variations in TA, although data are lacking in the present study to conclusively demonstrate this effect. Prior studies have shown that TA varies seasonally (Millero et al., 2001) and over diel cycles (present study; Yates et al., 2007) in response to fluctuations in calcification (Yates and Halley 2006) and salinity (net water balance), offering some explanation for the poor across-site relationship between $\Delta DO$ and $\Delta[H^+]$. TA generated by calcite dissolution or anaerobic biogeochemical processes like denitrification and $SO_4^{2-}$ reduction likely play an important, yet currently unknown role. Anaerobic generation of TA through denitrification or $SO_4^{2-}$ reduction in seagrass soils is an additional source not quantified here but should be addressed in the future. However, we can conclude that the observed lack of relationship between $\Delta DO$ and $\Delta[H^+]$ holds across the seagrass

productivity gradient in Florida Bay, indicating that this discrepancy between $NEP_{DO}$ and $NEP_{DIC}$ may extend across broad regions of the subtropics. This may challenge the application of new in-situ approaches that rely on variations in pH and DO alone to infer rates of biogeochemical processes (e.g. Long et al., 2015b).

Our results also suggest that the role of seagrass carbon cycling in larger, regional or global carbon cycles, may be much more complex than originally thought. Modern estimates of carbon uptake by seagrass ecosystems are based largely on measurements of $C_{org}$ burial rates or changes in standing stock of $C_{org}$ (Duarte et al., 2005; Fourqurean et al., 2012a; 2012b). While valuable, studies based solely on rates of $C_{org}$ burial integrate processes over long time scales, and may miss the impact of seagrass NEP and NEC on air-water $CO_2$ exchange and lateral $CO_{2(water)}$ and TA export. Indeed, it has been suggested that the dissolution of allochthonous carbonates in seagrass soils is an unrecognized sink of atmospheric $CO_2$ that exports TA to the coastal ocean on scales significant to global $CO_2$ budgets (Saderne et al 2019). If we are to more accurately constrain the role of seagrass ecosystems in the global carbon cycle, we must begin to consider the net ecosystem carbon balance (NECB), which is the residual carbon produced or consumed after all sources and sinks have been accounted for (Chapin et al., 2006). In aquatic systems, this will involve a precise measurement of the net ecosystem exchange (NEE) of $CO_2$ between the air and water. In the present study, we used a bulk-transfer equation (Eq 4 and 5) to estimate NEE, but new technologies such as eddy covariance and improved flux chambers mean that direct measurements of seagrass NEE are on the horizon. The combination of direct NEE measurements with rigorous assessments of NEP and NEC is one promising avenue through which NECB may be approached.

## 5. Conclusion

In this study, we present the first direct $NEP_{DIC}$ measurements in a representative seagrass meadow by combining rigorous carbonate system analysis with a diel sampling approach. We found negative $NEP_{DIC}$ and NEC at both sites, indicating that despite typical values of seagrass biomass and productivity (Table 1), both sites were net heterotrophic and net dissolving over the study period. When metabolic rates were low, they were likely affected by error due to the combined effect of advection and spatial concentration gradients, which can break the assumptions required for our 'open water' approach. On the contrary, this source of uncertainty was less important when metabolic rates were high. While we had some success in applying this 'open water' approach at these sites, we caution that error due to advection must be considered in sites where water currents are greater, or when the water depth is greater. Multiple lines of evidence point to sediment respiration and carbonate dissolution (Fig. 10) as drivers of negative NEP and NEC. While our isotopic and benthic flux measurements were coarse, they support the role of aerobic and anaerobic remineralization (denitrification and $SO_4^2$ reduction [Holmer et al., 2001; Eyre and Ferguson 2002; Smith et al., 2004]) coupled with carbonate dissolution (Jensen et al 1998, Burdige and Zimmerman 2002, Jensen et al 2009) as under-recognized components of total ecosystem NEP and NEC. Because of this, we express caution in interpreting our NEC results as strictly net production of $CaCO_3$; it appears that TA generated by anaerobic processes in the sediment likely influenced our estimates of NEC. Further studies should refine our estimates of benthic DIC and TA

fluxes from seagrass sediments (with benthic chambers [present study], underwater eddy covariance [Long et al., 2015b; Yamamoto et al., 2015], or pore-water modeling), and compare these values to other component fluxes of NEP and NEC (seagrass primary production, $CO_2$ flux, etc).

A key finding of this study was the divergence between $NEP_{DO}$ and $NEP_{DIC}$, which we attribute to the following factors 1) carbonate system buffering, which retains NEP-generated $CO_2$ in the water as DIC, 2) more rapid gas transfer, combined with a larger exchangeable pool for $O_2$ than for $CO_2$, and 3) a clear time-variant response of $NEP_{DO}$ to irradiance (Fig 7a). While DO-based approaches offer many advantages in cost and temporal coverage, we suggest that future studies should first constrain the underlying carbonate chemistry, and assess the relationship between $NEP_{DIC}$ and $NEP_{DO}$. Unfortunately, given the very limited temporal scope of this study, just 8 days, it is impossible to extend the results of this

study to longer time scales. At present, we cannot determine whether the seagrass ecosystem at this site is net dissolving and heterotrophic throughout the year, or even across seasons. More research is needed to assess the role of seasonal to annual scale variability in NEP and NEC on coastal ocean acidification trends. The use of new techniques, such as eddy covariance and improved autonomous instruments for pH, $pCO_2$, and TA, should allow future studies to build on this work and fill in our understanding of carbonate chemistry dynamics over longer, annual time scales. In particular, these new approaches should be

targeted at constraining NEE (air-water $CO_2$ exchange), in conjunction with direct and rigorous measurements of NEP and NEC. The combination of these approaches will allow for the first direct assessments of seagrass NECB, a critical next step in the valuation of seagrasses in the context of the global carbon cycle.

## Data Availability

All datasets generated during this project are published on the data sharing repository Figshare
(https://doi.org/10.6084/m9.figshare.7707029.v1). Further requests for data or methods sharing can be directed towards the corresponding author.

## Supplement

The supporting information related to this study will be published online.

## Author Contributions

BRV designed the research methodology and formal analysis for this study, while field and lab work was carried out by BRV and CL. Isotopic analysis of DIC was conducted by CO. The original draft of this manuscript was prepared by BRV, while further review and editing was conducted by JF, CO, and CL. We acknowledge the thoughtful comments and suggestions of three anonymous reviewers. Funding for this study was acquired by JF, and additionally through the DAAD (#57429828) from funds of the German Federal Ministry of Education and Research (BMBF).

**Competing Interests**

The authors declare no conflicts of interest

**Acknowledgements**

This work was supported by the US National Science Foundation through the Florida Coastal Everglades Long-Term
Ecological Research program under Grants No. DEB-1237517 and DEB-1832229. This project is also funded by the DAAD
(#57429828) from funds of the German Federal Ministry of Education and Research (BMBF). We thank Sara Wilson, Roxane
Bowden, Mary Zeller, and Mark Kershaw for assistance in the lab and field. We also appreciate the assistance of the National
Parks Service, who provided housing and lab space for this study. This is contribution #[updated upon publication] from the
Center for Coastal Oceans Research in the Institute of Water and Environment at Florida International University.

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
