# Peer review of "Net heterotrophy and carbonate dissolution in two subtropical seagrass meadows"

_Biogeosciences, 2019_

## Referee Comment (RC1) · Anonymous Referee #1 · 30 May 2019

**Summary:**

Van Dam et al. present seagrass metabolic rate estimates from two sites within Florida Bay. They found net heterotrophy and evidence for carbonate dissolution with the seagrass meadows and discuss the various drivers and implications of their metabolic rate findings for seagrass buffering of seawater chemistry. There is need for more information about seagrass metabolism and its relationships with water chemistry, so the study is well-motivated. The authors have clearly done a lot of work and I commend them for their effort. However, I have significant concerns about the metabolic rate calculations that constitute the main results of the paper.

I would not be comfortable seeing this paper published until the concerns are sufficiently addressed because I believe that addressing the concerns may change the main results of the paper.

In the first part of the review, I discuss my primary criticism of the study. I provide some detailed comments that pertain to the various sections, figures, and tables in the second part of the review. There is a short list of typos at the end of the review.

**Primary constructive criticism:**

The metabolic rate estimates are based on the "slack water" approach which considers an isolated pool of water such that changes in water chemistry cannot be attributed to advection or dispersion. Yet the authors do not sufficiently justify their adoption of the slack water simplification. These areas are not tidally isolated (e.g. tide pools), and although they feature low currents ($< 2$ cm/s; section 2.4), we have no sense of the spatial variation in O2 and DIC that would help us assess how any advective fluxes would compare to fluxes from gas exchange and/or metabolism. In particular, as highlighted by Lowe and Falter (2015), it is difficult to have both a) weak enough currents to minimize advective fluxes and b) strong enough turbulence to sufficiently mix the water column (see reference below).

I want to try to convince you that ignoring small spatial gradients and weak currents could cause you to misinterpret your metabolic rate data by ignoring advective fluxes. As an example, let's consider a simple advection-reaction model of the TA mass balance at one of the sites (equivalent to Eq. 1 in the paper with a term included for advection):

$$dTA/dt = u * delta\_TA/delta\_x - (2 * NEC/ rho * h)$$

At steady state ($dTA/dt = 0$), we could simplify this to:

$$(2 * NEC) / (rho * h * u) * delta\_x = delta\_TA$$

Assuming NEC = 5 mmol/m^2/hr (within the range of values presented in Fig. 4), rho = 1025 kg/m^3, u = 1 cm/s, h = 2m, and delta_x = 100m,  we can solve for delta_TA and

get:

delta_TA = 13 umol/kg

In other words, just a 13 umol/kg gradient between upstream and downstream TA in a 100m long meadow with a velocity of 1 cm/s (below your instrumental detection limit) would generate an advective flux equivalent to your reported rates of NEC. This TA range is far below your reported ranges in daily TA variability (which may be confounding temporal and spatial variability from advection). I suspect that your metabolic rates are really some combination of metabolism and advection. In some cases ignoring advection may be causing you to underestimate metabolism and in other cases may be causing you to overestimate metabolism.

*Without accounting for the role of advection in the TA, DIC, and O2 mass balances within the seagrass meadows, I am not confident in your conclusions about net heterotrophy and net dissolution.*

Given that the authors have O2 and pH measurements from some of the other FCE-LTER sites, they should explore how their metabolic rate estimates might change if they considered spatial variation in the biogeochemical parameters and associated advective fluxes (even if currents were < 2cm/s). They could at least put some error bounds on their metabolic rate estimates this way. Such an exercise would be especially doable if you have information on current direction from your tilt meters, even if you don't have current magnitude.

Finally, the authors implicitly acknowledge the role of advection when they discuss TA:DIC export (Fig. 9). The concept of export implies entry and exit flow through a system (in this case, each seagrass meadow), otherwise there would be no export. So how does one rationalize slack water metabolic rates and export at the same time?

Lowe, Ryan J., and James L. Falter. "Oceanic forcing of coral reefs." Annual review of marine science 7 (2015): 43-66.

**Detailed Comments**
**Methods:**
**2.1:** Move Table S1 to main text.
      Define "primary sites" here since you reference this phrase. Don't wait until 2.2 to define them.

**2.4:** Why such low accuracy on the pH sensors? SeaFETs are capable of accuracy approaching 0.01 pH units or better.

**2.6:** Why the poor precision on the DIC measurements? Please explain.

**2.7**:
Your NEC model does not account for changes in TA due to organic production, despite your acknowledgement in the text and Fig. 3 that TA is influenced by organic matter

production (see comment below about inconsistencies between delta_TA/delta_DIC ratios for organic production between your text and figure). You need to account for the other processes that influence TA in order to accurately calculate NEC.

Why are you using gas transfer velocity parameterizations designed for open ocean conditions when coastal parameterizations exist? See:

Ho, David T., et al. "Air-water gas exchange and CO2 flux in a mangrove-dominated estuary." Geophysical Research Letters 41.1 (2014): 108-113.
Ho, David T., et al. "Influence of current velocity and wind speed on air-water gas exchange in a mangrove estuary." Geophysical Research Letters 43.8 (2016): 3813-3821.

**Results:**

**3.1**

p. 7, L 17-18: The statement about lateral variations being insignificant because observed changes in SSS of < 1 is only correct if you knew that large spatial gradients in SSS existed and that they were correlated with TA, DIC, etc.

p. 7, L 22-23: Present O2 concentrations, not just percent of saturation (which is temperature and salinity dependent)

p. 7, L 28-29: t-tests assume independence between data sets, but your CO2 fluxes are likely to be linearly related (since the only difference is the estimated value of the gas piston velocity). I don't think t-tests are relevant since differences in gas flux should simply reflect differences in piston velocity.

p. 9, L 6-9: When you plot nTA against nDIC, the slope is not nTA:nDIC, but delta_nTA/delta_nDIC. Please be careful how you describe this in the text.

p. 9, L 9-10: When you only have two variables (nTA and nDIC), you can only resolve two processes (production and calcification). Right now, you are trying to resolve four processes (production, calcification, sulfate reduction, and denitrification) with only two variables. Your system is underdetermined.

**3.2**
p. 9, L 21: I do not believe this section is well served by the inclusion of metabolic rate comparisons between this study and previous seagrass metabolism studies. Move the comparisons to the paper Discussion.

p. 10, L 15-16: This is not the presentation of a statistical test result

**Figures**

**Fig 2:** I find this figure very difficult to follow. Multiple data series and and multiple variables along each subplot make it difficult to track what's going on where. Some axes are labeled and some are not. Please consider making additional plots, each with one variable, and labeling all axes. If there are too many resulting plots, you can put some in the supplement.

**Fig. 2g,h:** Point plots are difficult to track for understanding daily cycles. Recommend connecting points with a line.

**Fig. 3:** Where do you get the information that TA will *decrease* as DIC *decreases*? You reference the classical assumption of slight increases in TA with DIC uptake (p. 7, L 14 and also p. 17, L 16), but you have a positive line in Fig. 3 for TA/DIC relationships for organic production in Fig. 3 and the caption states "…., which generates 0.15 moles of TA for every mole of DIC respired." These two messages are inconsistent. Please clarify.

**Fig. 4:** Same comment as for Fig. 2 about multiple data sets and multiple variables. It is unnecessarily confusing to try to interpret these graphs and impatient readers won't invest much time and energy into attempting to do so. Also, same comment about connecting points with lines as with Fig. 2g,h. Please also provide a figure legend.

**Fig. 6:** Panels d) should be separated (split into a separate figure) from panels a-c) because they show fundamentally different relationships. Panels a-c) show relationships between metabolic rates and PAR. Panel d) shows relationships between oxygen and carbon fluxes during photosynthesis.

**Fig. 8:** Units on x-axis are incorrect. 1/DIC is in units of kg/umol, not umol/kg

**Fig. 9:** TA:DIC, not DIC:TA (check all labels)

**Tables:**

The information in Table S1 is key to understanding the differences between the high density and low density sites. At least an abridged version belongs in the main text.

**Typos:**

p. 2, L 22: Missing "it" between "While" and "is"

p. 7, L 2: Missing a space between "k600" and "parameterizations"

Manzello et al. (2012) reference (not "Manzanello), also correct in-line citation (p. 16, L 23)

---

## Short Comment (SC1) · 3 Jun 2019

I would like to personally thank this anonymous reviewer for their detailed comments on our discussion paper. This reviewer has clearly articulated a set of key points that we will need to address to move forward, and has taken the extra (and much appreciated) step of providing specific suggestions on how we may address these concerns. Again, I thank this reviewer for taking the time to read and fully consider our discussion paper, and hope they will be willing to again assess our work after our next set of revisions.

Thanks Bryce

---

## Referee Comment (RC2) · Anonymous Referee #2 · 9 Jun 2019

The study by Van Dam et al., aims at quantifying net primary production and calcification/dissolution rates of CaCO3 in Florida bay seagrass meadow. Although the methods used are correct, the study has a major flaw, and from my point of view, the manuscript in it's present form cannot be accepted. The authors are measuring benthic fluxes of TA, in seagrass and sediment, and consider that they are due to calcification or dissolution only (in seagrass, but not in sediment it seems). They therefore ignore all the other redox reactions producing of consuming TA, such as nitrification, denitrification, pyrite burial, sulfite burial, sulfate reduction etc. although those reactions are extremely important in seagrass beds, and indirectly controlled by the seagrass through sediment oxygenation and Corg addition. I strongly advise the authors to read Krumins et al., 2013 (biogeoscience) as well as Sippo et al., 2016 (global biogeochemical cycle). All the part regarding NEC is ill founded. The semi-quantitative arguments proposed by the authors tend to prove that the TA comes from dissolution (TA/DIC ratio and isotopes) are not convincing and only proves that part of the TA only come from this source. Measurements of fluxes of Ca2+, by titration, are necessary to quantify NEC. All parts regarding NEC should be removed, and only consider TA fluxes. This is a valuable and much needed data, the article should be rewritten to focus on this. NEC calculations could be proposed in discussion but it will need a very carefull and thorough discussions on sediment processes emitting TA.

Moreover, the study cover only two periods of ∼5 days in October and November. This temporal coverage is not sufficient to obtain significant results. More campaigns in other seasons are needed.

Some specific comments: Introduction:

P2 : Please develop how calcification emits CO2.

P2: 4-6: the experiment conducted by enriquez et al., consist in enclosing a piece of seagrass in a very small volume of water exposed to light. This is by no mean a proof that spontaneous CaCO3 can occur in the field. Besides, from my point of view, the observation of calcification within the tissues of seagrass they did remain to be confirmed.

P2: 34 - 35 : I do not understand that sentence

P4: 20. I don't find Karlsson et al., 2017 in the references.

P5: 11. Did you sampled discrete sample for spectrophotometric pH used for the seafet data validation? See Bresnahan et al., 2014 for example

P5: 25. Why using chamber for the bare sediment (and only for the sediment)

P6: 1-10: Did you used "Dickson" CRM

P6: 19: Please use the salinity normalization by Friis et al., 2003.

P7: 10-14. Please precise the dissociation constants used and evaluate the propagation of error on the CO2 calculated, using the fct error in seacarb. Please therefore take this error in consideration in subsequent calculations.

P7:11. Why not the latest Schmidt number calculations from Wanninkhof 2014? Please see Sippo et al, 2014.

P8: 7. Please express the hours in mean solar time. Fig 4, same.

P15: 9-12. I do not understand this section. The NEP(DIC) you calculate is a production rate of DIC, corrected for air-sea fluxes of CO2 and calcification (presumably), what is a proper way of doing. It is therefore including the DIC species HCO3- and CO32-, how can they escape the calculation?

P15: 26. Seagrass themselves? See earlier comment on Enriquez et al., 2014.

P15: 16. Your endvalues are far from 0 and close to the range for seagrass Corg. This does not reinforce the argument of TA coming from dissolution.

P16: 17. All your measurements are benthic TA fluxes. When it comes from bare sediment, it is a TA flux and when it comes from the seagrass, it is NEC.

P16: 20. Precisely, and denitrification and sulfate reduction emit TA and is NOT dissolution of CaCO3.

522. yes, exactly.

All the 4.3 section is dispensable.

―――――――――――――――――

---

## Referee Comment (RC3) · Anonymous Referee #3 · 10 Jun 2019

General comments

Van Dam et al. present short-term carbonate chemistry variability from two seagrass meadows in Florida Bay. Assessments of net ecosystem productivity (NEP) and net ecosystem calcification (NEC) indicated net heterotrophy and CaCO3 dissolution during eight days in the fall season. Furthermore, the authors compare NEP inferred from dissolved inorganic carbon measurements and oxygen measurements, and discuss reasons for and implications of the observed discrepancy. The study is well-designed and very timely as there is a lack of knowledge on how seagrass systems modify seawater carbonate chemistry on different temporal and spatial scales. However, although the carbonate chemistry methodology is appropriate, the interpretations and conclusions on TA fluxes and NEC would have benefited from additional measurements of

e.g., $Ca_{2+}$ and $SO_{4+}$. Without constraining other biogeochemical processes that affect DIC and TA, it should be more clearly indicated that some of the conclusions are associated with uncertainty and are speculative. Provided that the issues raised here are properly addressed, I would be happy to recommend this manuscript for publication. Please see my comments below.

The Methods section needs improvement. Information is missing on how several variables were measured and what sample sizes were used. Moreover, there is no information on how error propagation was calculated for your flux measurements, which could affect your conclusions. In section 2.1 and 2.2, how do you define your High Density and Low Density sites? Is it based on seagrass shoot density? If so, some quantification of this density would be beneficial for the justification of your site categorization. Above- and belowground biomass and productivity are reported for the two sites in Table S1, but it is unclear if your site categorization is based on any of these variables. Please state this clearly in the Methods section.

The Results section contains speculations and comparisons to previous studies that would be more suitable in the Discussion section. For example, p. 9, line 7-10, line 21; p. 10, line 1-7, line 19-20.

The Discussion section is well-written and easy to follow. However, I am missing some discussion on residence time within your two sites. You state that current flows were low, but no information is provided on tidal regime, prevailing wind direction etc. You briefly state in section 2.4 that current speeds were low (<2 cm s-1), but it is unclear if this means that you treat your sites as closed systems. If not, your budget in Section 4.3 neglects lateral import of DIC and TA from upstream systems as the export flux calculations are based on several assumptions that cannot be resolved with discrete point measurements of only DIC and TA. Aside from this, Section 4.3 brings up very important and relevant considerations for seagrass carbon cycling.

Specific comments

Abstract and Introduction

p. 1, line 10: This is purely semantic but I do not agree that the two seagrass meadows are contrasting. They are the same species, similar physicochemical conditions, similar productivity and water depth (Table S1).

p. 2, line 28: Seagrass beds and seagrass meadows are used interchangeably. Please use consistent terminology or if you treat these terms differently, please provide an explanation.

Methods

p. 3, line 23-24: Does "aboveground net primary productivity" refer to the data on row three in Table S1? If so, can you really say that they differed with such high and overlapping standard deviations ($2.05 \pm 0.90$ vs. $1.42 \pm 1.25$)? Were any statistical tests done to test these differences?

p. 4, line 5: Information on how many of the variables presented in Table S1 were measured is missing. For example, how many samples were taken to assess above- and belowground biomass? If only one sample per site was taken, I would be careful to state that they differed in biomass. Similarly, how were sediment carbon and nutrient contents measured. Are the reported C:N:P ratios on mass or molar basis?

p. 4, line 14-15: This is a bit confusing. Do these dates refer to the measurements of DOC, DIC, and TA for NEPDO, NEPDIC, and NEC or do they refer to air-water gas exchange? If the former, I suggest moving this last sentence up a bit or into the next paragraph where you describe the sampling campaigns.

p. 5, line 5: Is saturation state with respect to aragonite not relevant?

p. 6, line 1-7: Information on the accuracy of your measurements of DIC and TA is missing. Did you verify your measurements against Certified Reference Material? If you did, please state batch number. The precision of $\pm 5.11$ $\mu$mol kg-1 is quite poor. Could you provide a possible explanation for this? Were the DIC samples sufficiently

preserved (e.g., enough HgCl2)? Also, please add number of samples (n=) for your accuracy and precision assessments.

p. 7, line 6-8: What is the unit of k600? cm hr-1?

p. 7, line 10: End of sentence is missing.

Results

p. 7, line 17-20: This paragraph is a bit confusing as to what refers to the variation within each deployment and what refers to variation between each field campaign. I would not state that a salinity range from 31.45 to 34.67 is stable, but rather a substantial increase.

p. 7, line 23-24: You have already abbreviated your site names as HD and LD. Please be consistent with site terminology or remove the site abbreviation entirely (HD and LD) as there are already many other abbreviations throughout the manuscript.

p. 7, line 23: Please provide DO concentrations instead of just percent.

p. 9, line 9: These referenced studies did not measure sulfate reduction or denitrification. Please add additional references to back up the statement.

p. 10, line 5: Yes, but see Hines and Lyons 1982 and Holmer and Nielsen 1997.

p. 14, line 14-15: Although this is probably correct, I do not think that the observation of high benthic TA fluxes at the bare site necessarily means that sediment redox processes are not important for NEC. Furthermore, although sulfate reduction rates have been found to be higher in seagrass sediments, the oxygen release from seagrass roots can also lead to rapid re-oxidation of sulfide (consuming 1 mol TA).

Hines ME, Lyons WB (1982) Biogeochemistry of nearshore Bermuda sediments. I. Sulfate reduction rates and nutrient generation. Mar Ecol-Prog Ser:87-94

Holmer M, Nielsen SL (1997) Sediment sulfur dynamics related to biomass-density

patterns in Zostera marina (eelgrass) beds. Mar Ecol-Prog Ser 146:163-171

Discussion and Conclusion

p. 15, line 2: I suggest you include these productivity numbers in the Results section and also present the high variability (stdev of $\pm 0.9$ and $\pm 1.25$ $\mu$mol m-2 hr-1).

p.15, line 5: Do you consider seagrass belowground productivity as part of the "sediment processes"?

p. 16, line 16-18: Were these benthic chambers placed at bare spots within each seagrass meadow or at an adjacent bare site? Porewater chemistry vary on small spatial scales and can be quite different between unvegetated sediments and within the rhizosphere (e.g., due to differences in bioturbation, Corg, O2 release from roots etc.) and if your chamber measurements and $\delta$13C measurements are spatially decoupled I would not combine the two as aggregate evidence.

p. 16, line 19-21: Yes, but these processes (along with other redox processes) could also affect your NEC estimates. Your TA:DIC ratios are the result of a combination of these processes and without measuring any other reactants and products it is difficult to constrain their contribution to your TA flux. Additionally, organic alkalinity may be produced in the sediments which is not accounted for in TA (see e.g., Lukawska-Matuszewska, 2016).

p. 16, line 21-24: Yes, indeed. Very well formulated.

p. 17, 2-3: I suggest that these reflections are included in the abstract as well.

p. 17, line 10: . . . or throughout the year.

p. 18, line 23-24: Very true, but Corg burial operates on much longer timescales than the diel (fall season) NEP and NEC measured in this study.

Lukawska-Matuszewska K (2016). Contribution of non-carbonate inorganic and organic alkalinity to total measured alkalinity in pore waters in marine sediments (Gulf of

Gdansk, S-E Baltic Sea). Marine Chemistry 186:211-220

Figures

Figure 1 and 2: Please define in the Methods section or figure caption what U10 represents, to help readers who are not familiar with wind speed terminology.

Figure 2: Please place panel letters (a-g) so that they do not interfere with data points.

Figure 2g-h: Please use same nTA y-axis range for both campaigns to allow for easier comparison. Following these time series would also be easier if you use lines to connect data points.

Figure 3: Why do you not include the slopes for sulfate reduction and denitrification as you mention these processes in p. 9, line 9-10?

Figure 7: This figure is quite confusing to me. The generalized pattern in PPR, [P] and TA is unclear. Does it refer to the sites on the map (e.g., PPR and [P] decreases eastward, TA is high in site BA but low in sites SB, HD and LD?). Please clarify in the figure caption.

Figure 8: I suggest you move the legend from the inset figure to the main figure and increase the font size. Also, try and increase the size of the dotted confidence interval lines as these are very difficult to see.

Figure 9: Change "DIC:TA" to "TA:DIC".

Technical corrections

p. 2, line 23: Insert "it" after "while"

p. 2, line 30: Change "seagrasses meadows" to "seagrass meadows".

p. 3, line 9-10: Is there a word missing in this sentence? E.g. [. . .], suggesting the "significant/important/negligible" role of NEC or anaerobic catabolic processes in generating excess CO2.

p. 3, line 11-14: Many "potential" in this paragraph. I suggest you remove "potential" from the sentence "discuss potential differences"

p. 5, line 10: Superscript "-1" in mg L-1 and % saturation)

p. 9, line 6: Missing an "and" before "calcification".

p. 10, line 10: Should it not be "[. . .] sampling campaign 1 (a,b) and 2 (c,d)"?

p. 16, line 16: Change NEPDIC to NEPDIC.

p. 19, line 2: I do not think coastal Ocean is spelled with a capital O.

p. 19, line 29: Remove "of pH".

---

## Author Comment (AC1) · 22 Jul 2019

We appreciate Reviewer 1's thorough comments and constructive criticism of our manuscript. Their primary critique is well-founded and articulated, and we are thankful for the detailed argument that they have laid out in their review. Indeed, flow in seagrass systems is complex, producing vertical heterogeneities in water column physical and chemical properties. For example, flow is significantly reduced in the canopy, increasing the residence time of water in the canopy relative to the overlying water (Peterson et al, 2004). Shear between these two compartments (in and out of the canopy) drives vertical exchange across the canopy interface that partially or wholly homogenizes water chemistry. At a smaller scale, this turbulent mixing also helps to alleviate carbon limitation that may build up in the seagrass blade boundary layer (Koch 1994). Our NEP/NEC estimates were derived from concentrations measured near the surface. These measurements represent the cumulative effect of lateral DIC/TA fluxes (which we assume to be minor) and turbulent/diffusive exchange between the seagrass canopy and overlying water (which we assume are dominant). This is a partial motivation for why we chose surface-water rather than within-canopy measurements, because it integrates the seagrass metabolic signal from a larger footprint. Still, there is a potential for our slack water approach to be biased by lateral water exchanges, which we will try to address to the best of our ability here.

Unfortunately, we don't have any empirical data specifically addressing the spatial variability of carbonate chemistry at these sites, but we can build one line of evidence from the data that we do have. Our two sites are separated by a linear distance of approximately 4 km. Looking at figure 2, we can approximate the difference in nTA and nDIC between the sites to be at most 300 $\mu$mol/kg. Hence, we have an approximate spatial gradient of at most 75 $\mu$mol/kg/km (300 $\mu$mol/kg / 4km). This corresponds to at most 7.5 $\mu$mol/kg over a 100m stretch, which is about half of the 13 $\mu$mol/kg estimate that reviewer 1 derives in their comments. Furthermore, our seagrass meadows are much larger than 100m, in fact are typically a factor of ~5x greater (>0.5 km$^2$). Hence, the comparable TA gradient required to explain our metabolic fluxes would be appreciably greater, on the order of ~65 $\mu$mol/kg.

As further evidence, we are including the following figure which shows current speed and direction from the tilt current meters (TCMs). From this, it appears that flow was not unidirectional at these sites over the study period, but was instead variable in direction without a clear mode which might suggest tidal or wind-seiche. While we are reluctant to use these data in our manuscript because the water velocities were below the detection limit of the TCM, we hope they offer some support to our argument in this discussion forum. One prior study at a site just west of ours also reported generally low water current, especially within the seagrass canopy, despite a slightly greater tidal influence there (Hansen et al., 2017). Hence, we feel confident that water current at our site was indeed low. We also see no clear link between current direction and changes in TA/DIC, which should be apparent if there was a distinct TA/DIC source whose signature was being advected over our site. For example, there was a subtle decrease in salinity of ~0.5 on the morning of 11/27 at the high-density site (Fig. 2), but the indicated current speed and direction were apparently consistent during this time period (attached figure). Lastly, it is possible that small inputs of fresh water, either through surface or groundwater channels may have significant and nonlinear impacts on carbonate chemistry. However, the attached scatter plot of salinity vs depth indicates that the small changes in water level we observed did not coincide with any clear changes in salinity (i.e. freshwater input).

So, we strongly agree that the combination of spatial variability in carbonate chemistry and advection can cause TA/DIC variability that may impact the ability to estimate NEM/NEC. This would be especially problematic if we had collected water samples *within* the seagrass canopy where water chemistry is much more variable in space/time. However, if we consider all of these lines of evidence, along with the fact that our measurements were made above the canopy, we argue that lateral mixing over the study period was likely relatively low, and likely not sufficient to drive the diel variations we observed, which were generally 50-100 $\mu$mol/kg.

In light of Reviewer 1's concerns regarding the assumptions involved in the TA/DIC budget used for Figure 9, we have elected to remove section 4.3 (TA/DIC export) and figure 9 from the manuscript. Furthermore, we have made a concerted effort to more clearly state the assumptions and limitations of our 'slack water' approach throughout the manuscript.

Peterson, C & Luettich, Jr, R & Micheli, F & Skilleter, G. (2004). Attenuation of water flow inside seagrass canopies of differing structure. Marine Ecology Progress Series. 268.

Hansen, J. C. R., & Reidenbach, M. A. (2017). Turbulent mixing and fluid transport within Florida Bay seagrass meadows. Advances in Water Resources, 108, 205–215. doi:10.1016/j.advwatres.2017.08.001

Koch, E.W. Marine Biology (1994) 118: 767. https://doi.org/10.1007/BF00347527 10.3354/meps268081.

[Figure]

Detailed comments:

Methods:

2.1: Move Table S1 to main text.

Table S1 moved to the main text as table 1

Define "primary sites" here since you reference this phrase. Don't wait until 2.2 to define them.

This term is now introduced in the first sentence of 2.1

2.4: Why such low accuracy on the pH sensors? SeaFETs are capable of accuracy approaching 0.01 pH units or better.

This is the accuracy listed on the manufacturer's website (https://www.seabird.com/seafet-v2-ocean-ph-sensor/product-details?id=54627921732). The precision is indeed much better than 0.05.

2.6: Why the poor precision on the DIC measurements? Please explain.

While TA was analyzed on a commercial instrument, we did not have such a machine for DIC determination. Instead, our DIC measurements were made on a home-made analyzer which consisted of a small impinger filled with 10% HCl, an N2 carrier gas, and a bench-top IRGA

(Licor 6262). There was uncertainty in sample injection, which was done manually, and peak area integration, which was done by the IRGA. While our precision was lower for DIC than for TA, it was still reasonably close to what is achieved by commercial units, which typically achieve ~2 $\mu$mol/kg accuracy (e.g. Apollo SciTech ASC3 [http://www.apolloscitech.com/dic.html]). While other instruments like the VINDTA 3C (http://www.marianda.com/index.php?site=products&subsite=vindta3c) claim ~1 $\mu$mol/kg precision, reported standard deviations of CRMs are generally higher for both TA and DIC, closer to 2-4 $\mu$mol/kg (McMahon et al., 2018; Lemay et al., 2018; Turk et al, 2016, etc…).

McMahon, A., I. R. Santos, K. G. Schulz, T. Cyronak, and D. T. Maher. 2018. Determining coral reef calcification and primary production using automated alkalinity, pH and p CO 2 measurements at high temporal resolution. Estuar. Coast. Shelf Sci. **209**: 80–88. doi:10.1016/j.ecss.2018.04.041

Lemay, J., H. Thomas, S. E. Craig, W. J. Burt, K. Fennel, and B. J. W. Greenan. 2018. Hurricane Arthur and its effect on the short-term variability of p CO 2 on the Scotian Shelf , NW Atlantic. Biogeosciences 2111–2123.

Turk, D., J. M. Bedard, W. J. Burt, and others. 2016. Estuarine , Coastal and Shelf Science Inorganic carbon in a high latitude estuary-fjord system in Canada ' s eastern Arctic. Estuar. Coast. Shelf Sci. 178: 137–147. doi:10.1016/j.ecss.2016.06.006

2.7:Your NEC model does not account for changes in TA due to organic production, despite your acknowledgement in the text and Fig. 3 that TA is influenced by organic matter production (see comment below about inconsistencies between delta_TA/delta_DIC ratios for organic production between your text and figure). You need to account for the other processes that influence TA in order to accurately calculate NEC.
Why are you using gas transfer velocity parameterizations designed for open ocean conditions when coastal parameterizations exist? See:
Ho, David T., et al. "Air-water gas exchange and CO2 flux in a mangrove-dominated estuary." Geophysical Research Letters 41.1 (2014): 108-113.
Ho, David T., et al. "Influence of current velocity and wind speed on air-water gas exchange in a mangrove estuary." Geophysical Research Letters 43.8 (2016): 3813-3821.

We chose to apply two separate parameterizations because together they constitute what might be considered a maximum range in $k$, within which we expect that the actual value lies. Because calculated NEP using these two (excessively) different parameterizations were very similar, we felt justified in reporting a single value from Ho 2006. While we are well aware of the Ho 2016 and Ho 2014 parameterizations, we elected not to use them because of the lack of quality water velocity data, and the fact that currents at our site (likely < 2 cm/s) were at least an order of magnitude lower than the velocities in the tidal river in Ho 2016 (20-40 cm/s). Likewise, Ho et al., 2014 reports average tidal velocities of ~35 cm/s, well outside the range at our site.

Results:

3.1

p. 7, L 17-18: The statement about lateral variations being insignificant because observed changes in SSS of < 1 is only correct if you knew that large spatial gradients in SSS existed and that they were correlated with TA, DIC, etc.

This is a very good point. We have removed 'lateral mixing' from the sentence, and have clarified that we were referring to sources of fresh water, not TA or DIC.

p. 7, L 22-23: Present O2 concentrations, not just percent of saturation (which is temperature and salinity dependent)

DO is now presented as a concentration rather than a percent saturation (Fig 2), and the text references have been corrected as well. The diel trends in DO remain apparent in the figure.

p. 7, L 28-29: t-tests assume independence between data sets, but your CO2 fluxes are likely to be linearly related (since the only difference is the estimated value of the gas piston velocity). I don't think t-tests are relevant since differences in gas flux should simply reflect differences in piston velocity.

We have removed the discussion of CO2 flux t-tests from section 3.

p. 9, L 6-9: When you plot nTA against nDIC, the slope is not nTA:nDIC, but delta_nTA/ delta_nDIC. Please be careful how you describe this in the text.

We have added a brief clarification on this point.

p. 9, L 9-10: When you only have two variables (nTA and nDIC), you can only resolve two processes (production and calcification). Right now, you are trying to resolve four processes (production, calcification, sulfate reduction, and denitrification) with only two variables. Your system is underdetermined.

We very much agree, and have added a sentence at the end of the paragraph reiterating this point.

3.2p. 9, L 21: I do not believe this section is well served by the inclusion of metabolic rate comparisons between this study and previous seagrass metabolism studies. Move the comparisons to the paper Discussion.

Yes, this discussion of metabolic rates in the context of previous studies is not suited for the results section. It has been moved to the discussion section 4.1.

p. 10, L 15-16: This is not the presentation of a statistical test result

These sentences were removed as per Reviewer 1's earlier comments.

Figures

Fig 2: I find this figure very difficult to follow. Multiple data series and and multiple variables along each subplot make it difficult to track what's going on where. Some axes are labeled and some are not. Please consider making additional plots, each with one variable, and labeling all axes. If there are too many resulting plots, you can put some in the supplement.

Fig. 2g,h: Point plots are difficult to track for understanding daily cycles. Recommend connecting points with a line.

We appreciate the advice, and have revised figure 2 to include axis titles for all sub-figures and have connected the points in figures g and h with lines.

Fig. 3: Where do you get the information that TA will decrease as DIC decreases? You reference the classical assumption of slight increases in TA with DIC uptake (p. 7, L 14 and also p. 17, L 16), but you have a positive line in Fig. 3 for TA/DIC relationships for organic production in Fig.

3 and the caption states "...., which generates 0.15 moles of TA for every mole of DIC respired." These two messages are inconsistent. Please clarify.

Reviewer 1 is correct, we should present a slope of -0.15 for the blue line in figure 3 representing TA uptake with productivity on $NO_3$. This has been corrected in the new figure.

Fig. 4: Same comment as for Fig. 2 about multiple data sets and multiple variables. It is unnecessarily confusing to try to interpret these graphs and impatient readers won't invest much time and energy into attempting to do so. Also, same comment about connecting points with lines as with Fig. 2g,h. Please also provide a figure legend.

We regret that this figure is difficult to follow, but we have tried a number of ways to plot these data and settled on the current display as the least bad representation. On a previous version of this figure, we tried to connect the points with lines, but it became far too busy and difficult to see. We also tried to use box and violin plots, but there simply aren't enough data points to make these plots work.

Fig. 6: Panels d) should be separated (split into a separate figure) from panels a-c) because they show fundamentally different relationships. Panels a-c) show relationships between metabolic rates and PAR. Panel d) shows relationships between oxygen and carbon fluxes during photosynthesis.

We appreciate Reviewer 1's advice, and have split Figure 6 into two separate figures. The in-text references have been revised accordingly.

Fig. 8: Units on x-axis are incorrect. 1/DIC is in units of kg/umol, not umol/kg

Units have been corrected in Figure 8.

Fig. 9: TA:DIC, not DIC:TA (check all labels)

Figure 9 has been removed from the manuscript.

Tables:

The information in Table S1 is key to understanding the differences between the high density and low density sites. At least an abridged version belongs in the main text.

Table S1 has been moved to the main text.

Typos:

p. 2, L 22: Missing "it" between "While" and "is"

p. 7, L 2: Missing a space between "k600" and "parameterizations"

Manzello et al. (2012) reference (not "Manzanello), also correct in-line citation (p. 16, L 23)

We thank Reviewer 1 for catching these mistakes, which we have now corrected.

---

## Author Comment (AC2) · 22 Jul 2019

The study by Van Dam et al., aims at quantifying net primary production and calcification/dissolution rates of CaCO3 in Florida bay seagrass meadow. Although the methods used are correct, the study has a major flaw, and from my point of view, the manuscript in it's present form cannot be accepted. The authors are measuring benthic fluxes of TA, in seagrass and sediment, and consider that they are due to calcification or dissolution only (in seagrass, but not in sediment it seems). They therefore ignore all the other redox reactions producing of consuming TA, such as nitrification, denitrification, pyrite burial, sulfite burial, sulfate reduction etc. although those reactions are extremely important in seagrass beds, and indirectly controlled by the seagrass through sediment oxygenation and Corg addition. I strongly advise the authors to read Krumins et al., 2013 (biogeoscience) as well as Sippo et al., 2016 (global biogeochemical cycle). All the part regarding NEC is ill founded. The semi-quantitative arguments proposed by the authors tend to prove that the TA comes from dissolution (TA/DIC ratio and isotopes) are not convincing and only proves that part of the TA only come from this source. Measurements of fluxes of Ca2+, by titration, are necessary to quantify NEC. All parts regarding NEC should be removed, and only consider TA fluxes. This is a valuable and much needed data, the article should be rewritten to focus on this. NEC calculations could be proposed in discussion but it will need a very carefull and thorough discussions on sediment processes emitting TA.

Moreover, the study cover only two periods of ~5 days in October and November. This temporal coverage is not sufficient to obtain significant results. More campaigns in other seasons are needed.

We appreciate reviewer 2's constructive criticism, and have made a concerted effort to address their concerns regarding the role of anaerobic processes on NEC. Throughout the manuscript, we have added text reminding the reader when specific results may have been affected by anaerobic TA generation. We have also included extra text throughout that emphasizes the limited temporal scope of the study, and expressed the need for future studies using different approaches over longer time scales in order to confirm or refute our findings. We hope that these changes, along with those that have been made following reviewer 1 and 3's suggestions will be satisfactory for this reviewer.

Some specific comments: Introduction:

P2 : Please develop how calcification emits CO2.

This sentence was expanded to clarify how calcification generates CO2.

P2: 4-6: the experiment conducted by enriquez et al., consist in enclosing a piece of seagrass in a very small volume of water exposed to light. This is by no mean a proof that spontaneous CaCO3 can occur in the field. Besides, from my point of view, the observation of calcification within the tissues of seagrass they did remain to be confirmed.

We agree with the reviewer that more studies are required to confirm that CaCO3 formation occurs within seagrass tissues and have added phrasing to reiterate this point here.

P2: 34 - 35 : I do not understand that sentence

This sentence was revised to clarify that seagrasses can affect local pH trends by consuming DIC that was generated in adjacent mangroves.

P4: 20. I don't find Karlsson et al., 2017 in the references.

We apologize for the omission; this citation is now included in the reference list.

P5: 11. Did you sampled discrete sample for spectrophotometric pH used for the seafet data validation? See Bresnahan et al., 2014 for example

These SeaFET data were not used to calculate DIC/TA for metabolism assessments, and were simply presented to show the large diel cycles in pH. Our original intent was to estimate NEP/NEC at higher temporal resolution using sensor pH and pCO2 data, but because we were not confident in the pCO2 data, we could not do so.

P5: 25. Why using chamber for the bare sediment (and only for the sediment)

The intent here was to isolate the sediment source of TA/DIC by excluding seagrass aboveground biomass, thereby excluding any consumption or production by seagrass aboveground shoots themselves. We have edited this sentence to clarify the point.

P6: 1-10: Did you used "Dickson" CRM
Yes, and this is now explained in greater detail.

P6: 19: Please use the salinity normalization by Friis et al., 2003.

The water budget of Florida Bay is dominated by exchange with the ocean and evaporation and precipitation, which are approximately a factor of 10 greater than surface water inputs which may have a non-zero TA/DIC endmember (Nuttle et al., 2000). Therefore, we believe that the most appropriate approach is to normalize TA and DIC using a zero-salinity endmember, which represents the effect of precipitation and evaporation. Furthermore, the small freshwater input that does enter the northern bay through shark river slough has a highly variable TA concentration, and is located a great distance from our study sites.

P7: 10-14. Please precise the dissociation constants used and evaluate the propagation of error on the $CO_2$ calculated, using the fct error in seacarb. Please therefore take this error in consideration in subsequent calculations.

We are confident that the largest source of error in our $CO_2$ flux determination is derived from our parameterization of gas transfer, which is why we used two different equations to estimate k600. Furthermore, $CO_2$ flux represents only a very small fraction (median=1.3%) of the estimated NEP rates. Therefore, we feel confident in presenting the results using a single $H_2CO_3$ dissociation constant.

P7:11. Why not the latest Schmidt number calculations from Wanninkhof 2014? Please see Sippo et al, 2014.

We regret not using the updated Sc values from Wanninkhof 2014 in our analysis. However, re-doing the entire analysis with the 2014 values would require significant time, and would not appreciably change our $CO_2$ flux estimates, which are most sensitive to variations in the gas transfer velocity (k600), rather than variations in Sc which are small. If reviewer 2 deems it necessary that we re-calculate all metabolism estimates with the updated $CO_2$ and $O_2$ Sc values from Wanninkhof 2014, we would of course be willing to do so.

P8: 7. Please express the hours in mean solar time. Fig 4, same.

Time is expressed in local time (EDT or EST) throughout the rest of the manuscript, so we elect to present time in the same format in this figure to avoid confusion.

P15: 9-12. I do not understand this section. The NEP(DIC) you calculate is a production rate of DIC, corrected for air-sea fluxes of $CO_2$ and calcification (presumably), what is a proper way of doing. It is therefore including the DIC species $HCO_3^-$ and $CO_3^{2-}$, how can they escape the calculation?

This section is not intended to say that DIC is somehow 'escaping' the NEP calculation, rather that the large pool of DIC makes NEP calculated with DIC less sensitive to variations in gas transfer than NEP calculated with $O_2$.

P15: 26. Seagrass themselves? See earlier comment on Enriquez et al., 2014.

Yes. While there is debate over the extent to which seagrass internal calcification occurs, we have mentioned this previously in the manuscript (as per this reviewer's suggestion), and at this point, we also mention other calcifiers which likely contribute in some extent to our NEC estimates.

P15: 16. Your endvalues are far from 0 and close to the range for seagrass Corg. This does not reinforce the argument of TA coming from dissolution.

Indeed, the indicated y-intercept of the Keeling plot does suggest an endmember closer to seagrass Corg. However, the 95% confidence interval for the y-intercept is ~3-11 for the high-density site, and, ~2-16 for the low-density site. This factor, along with the extreme extrapolation involved, means that we cannot confidently say that the endmember is either decidedly "carbonate" or "seagrass OM".

P16: 17. All your measurements are benthic TA fluxes. When it comes from bare sediment, it is a TA flux and when it comes from the seagrass, it is NEC.

We have revised the previous sentence to clarify our intended message that sediment-water TA/DIC fluxes may at times explain a large fraction of measured NEC.

P16: 20. Precisely, and denitrification and sulfate reduction emit TA and is NOT dissolution of $CaCO_3$.

As per reviewer 2's comments, we have added a sentence expanding on the role of anaerobic processes on TA exchanges.

522. yes, exactly.

All the 4.3 section is dispensable.

As per all 3 reviewers suggestions, section 4.3 was significantly reduced in length and the budget was entirely removed. The remainder of section 4.3 received positive comments from the other reviewers, and we think that it brings up important points, so we elect to keep it in this revision.

---

## Author Comment (AC3) · 22 Jul 2019

The comment was uploaded in the form of a supplement:
https://www.biogeosciences-discuss.net/bg-2019-191/bg-2019-191-AC3-supplement.pdf
* * *

---

## Author Response (AR1)

**Summary:**

Van Dam et al. present seagrass metabolic rate estimates from two sites within Florida Bay. They found net heterotrophy and evidence for carbonate dissolution with the seagrass meadows and discuss the various drivers and implications of their metabolic rate findings for seagrass buffering of seawater chemistry. There is need for more information about seagrass metabolism and its relationships with water chemistry, so the study is well-motivated. The authors have clearly done a lot of work and I commend them for their effort. However, I have significant concerns about the metabolic rate calculations that constitute the main results of the paper.

I would not be comfortable seeing this paper published until the concerns are sufficiently addressed because I believe that addressing the concerns may change the main results of the paper.

In the first part of the review, I discuss my primary criticism of the study. I provide some detailed comments that pertain to the various sections, figures, and tables in the second part of the review. There is a short list of typos at the end of the review.

**Primary constructive criticism:**

The metabolic rate estimates are based on the "slack water" approach which considers an isolated pool of water such that changes in water chemistry cannot be attributed to advection or dispersion. Yet the authors do not sufficiently justify their adoption of the slack water simplification. These areas are not tidally isolated (e.g. tide pools), and although they feature low currents (< 2 cm/s; section 2.4), we have no sense of the spatial variation in O2 and DIC that would help us assess how any advective fluxes would compare to fluxes from gas exchange and/or metabolism. In particular, as highlighted by Lowe and Falter (2015), it is difficult to have both a) weak enough currents to minimize advective fluxes and b) strong enough turbulence to sufficiently mix the water column (see reference below).

I want to try to convince you that ignoring small spatial gradients and weak currents could cause you to misinterpret your metabolic rate data by ignoring advective fluxes. As an example, let's consider a simple advection-reaction model of the TA mass balance at one of the sites (equivalent to Eq. 1 in the paper with a term included for advection):

dTA/dt = u \* delta\_TA/delta\_x - (2 \* NEC/ rho \* h) At steady state (dTA/dt = 0), we could simplify this to: (2 \* NEC) / (rho \* h \* u) \* delta\_x = delta\_TA

Assuming NEC = 5 mmol/m2/hr (within the range of values presented in Fig. 4), rho = 1025 kg/m3, u = 1 cm/s, h = 2m, and delta\_x = 100m, we can solve for delta\_TA and get: delta\_TA = 13 umol/kg

In other words, just a 13 umol/kg gradient between upstream and downstream TA in a 100m long meadow with a velocity of 1 cm/s (below your instrumental detection limit) would generate an advective flux equivalent to your reported rates of NEC. This TA range is far below your reported ranges in daily TA variability (which may be confounding temporal and spatial variability from advection). I suspect that your metabolic rates are really some combination of metabolism and advection. In some cases ignoring advection may be causing you to underestimate metabolism and in other cases may be causing you to overestimate metabolism.

Without accounting for the role of advection in the TA, DIC, and O2 mass balances within the seagrass meadows, I am not confident in your conclusions about net heterotrophy and net dissolution.

Given that the authors have O2 and pH measurements from some of the other FCE- LTER sites, they should explore how their metabolic rate estimates might change if they considered spatial variation in the biogeochemical parameters and associated advective fluxes (even if currents were < 2cm/s). They could at least put some error bounds on their metabolic rate estimates this way. Such an exercise would be especially doable if you have information on current direction from your tilt meters, even if you don't have current magnitude.

Finally, the authors implicitly acknowledge the role of advection when they discuss TA:DIC export (Fig. 9). The concept of export implies entry and exit flow through a system (in this case, each seagrass meadow), otherwise there would be no export. So how does one rationalize slack water metabolic rates and export at the same time?

Lowe, Ryan J., and James L. Falter. "Oceanic forcing of coral reefs." Annual review of marine science 7 (2015): 43-66.

We appreciate Reviewer 1's thorough comments and constructive criticism of our manuscript. Their primary critique is well-founded and articulated, and we are thankful for the detailed argument that they have laid out in their review. Indeed, flow in seagrass systems is complex, producing vertical heterogeneities in water column physical and chemical properties. For example, flow is significantly reduced in the canopy, increasing the residence time of water in the canopy relative to the overlying water (Peterson et al, 2004). Shear between these two compartments (in and out of the canopy) drives vertical exchange across the canopy interface that partially or wholly homogenizes water chemistry. At a smaller scale, this turbulent mixing also helps to alleviate carbon limitation that may build up in the seagrass blade boundary layer (Koch 1994). Our NEP/NEC estimates were derived from concentrations measured near the surface. These measurements represent the cumulative effect of lateral DIC/TA fluxes (which we assume to be minor) and turbulent/diffusive exchange between the seagrass canopy and overlying water (which we assume are dominant). This is a partial motivation for why we chose surface-water rather than within-canopy measurements, because it integrates the seagrass metabolic signal from a larger footprint. Still, there is a potential for our slack water approach to be biased by lateral water exchanges, which we will try to address to the best of our ability here.

Unfortunately, we don't have any empirical data specifically addressing the spatial variability of carbonate chemistry at these sites, but we can build one line of evidence from the data that we do have. Our two sites are separated by a linear distance of approximately 4 km. Looking at figure 2, we can approximate the difference in nTA and nDIC between the sites to be at most 300  $\mu$ mol/kg. Hence, we have an approximate spatial gradient of at most 75  $\mu$ mol/kg/km (300  $\mu$ mol/kg / 4km). This corresponds to at most 7.5  $\mu$ mol/kg over a 100m stretch, which is about half of the 13  $\mu$ mol/kg estimate that reviewer 1 derives in their comments. Furthermore, our seagrass meadows are much larger than 100m, in fact are typically a factor of ~5x greater (>0.5 km2). Hence, the comparable TA gradient required to explain our metabolic fluxes would be appreciably greater, on the order of ~65  $\mu$ mol/kg.

As further evidence, we are including the following figure which shows current speed and direction from the tilt current meters (TCMs). From this, it appears that flow was not unidirectional at these sites over the study period, but was instead variable in direction without a clear mode which might suggest tidal or wind-seiche. While we are reluctant to use these data in

our manuscript because the water velocities were below the detection limit of the TCM, we hope they offer some support to our argument in this discussion forum. One prior study at a site just west of ours also reported generally low water current, especially within the seagrass canopy, despite a slightly greater tidal influence there (Hansen et al., 2017). Hence, we feel confident that water current at our site was indeed low. We also see no clear link between current direction and changes in TA/DIC, which should be apparent if there was a distinct TA/DIC source whose signature was being advected over our site. For example, there was a subtle decrease in salinity of ~0.5 on the morning of 11/27 at the high-density site (Fig. 2), but the indicated current speed and direction were apparently consistent during this time period (attached figure). Lastly, it is possible that small inputs of fresh water, either through surface or groundwater channels may have significant and nonlinear impacts on carbonate chemistry. However, the attached scatter plot of salinity vs depth indicates that the small changes in water level we observed did not coincide with any clear changes in salinity (i.e. freshwater input).

So, we strongly agree that the combination of spatial variability in carbonate chemistry and advection can cause TA/DIC variability that may impact the ability to estimate NEM/NEC. This would be especially problematic if we had collected water samples *within* the seagrass canopy where water chemistry is much more variable in space/time. However, if we consider all of these lines of evidence, along with the fact that our measurements were made above the canopy, we argue that lateral mixing over the study period was likely relatively low, and likely not sufficient to drive the diel variations we observed, which were generally 50-100  $\mu$ mol/kg.

In light of Reviewer 1's concerns regarding the assumptions involved in the TA/DIC budget used for Figure 9, we have elected to remove section 4.3 (TA/DIC export) and figure 9 from the manuscript. Furthermore, we have made a concerted effort to more clearly state the assumptions and limitations of our 'slack water' approach throughout the manuscript.

Peterson, C & Luettich, Jr, R & Micheli, F & Skilleter, G. (2004). Attenuation of water flow inside seagrass canopies of differing structure. Marine Ecology Progress Series. 268.

Hansen, J. C. R., & Reidenbach, M. A. (2017). Turbulent mixing and fluid transport within Florida Bay seagrass meadows. Advances in Water Resources, 108, 205–215. doi:10.1016/j.advwatres.2017.08.001

Koch, E.W. Marine Biology (1994) 118: 767. https://doi.org/10.1007/BF00347527 10.3354/meps268081.

---

## Referee Report (RR1)

Van Dam et al. have put a lot of work into addressing the three very thorough reviews of their initial submission. In general, I think they did a good job addressing my constructive criticisms of the initial manuscript and it is much improved.

I appreciated Van Dam et al.'s response to my primary constructive criticism RE: missing advective terms in their metabolic rate models. They did a nice job diving back into their data to assess spatial gradients in TA and attempted to place them into context with their diel TA ranges to argue that advective fluxes can be safely ignored, after describing in the text. Below, I will show why I disagree with the assertion that these advective fluxes can be ignored. I will walk the authors through some examples and explain why I believe the authors should use these "missing" advective fluxes as error bounds on their metabolic rates.

In this example, we will continue to stick to NEC and TA fluxes, but the same concepts are analogous for DIC (and will need to be applied equally).

As the authors stated int their response, they observed spatial TA gradients of 75 umol/kg/km. Let's call this variable: dnTA/dx.

Now, observing NEC rates of +/- 15 mmol/m^2/hr (Fig. 6c), we can invert Eq. 1 to calculate the time rate of change of TA (dnTA/dt). Let's assume rho=1025 kg/m^3 and h = 2 meters. Furthermore, let's also do the calculation for NEC rates of 5, 10, and 15 mmol/m^2/hr (we will be sign-agnostic because it's the magnitudes we are most interested in).

So inverting Eq. 1 to solve for the dnTA/dt yields:

dnTA/dt = -2 * NEC / rho * h

And so for our three test cases of NEC = 5, 10, and 15 mmol/m^2/hr, dnTA/dt ~ 5, 10, and 15 umol/kg/hr (with some simplified rounding)

Now let's compare these estimates for the time-varying term against your spatial gradients. Your plots of TCMs suggest flow speeds below 1 cm/s (acknowledging that the limit of detection on the instrument is 2 cm/s). So let's consider two test cases of u= 0.5 cm/s and u = 1cm/s (I know the displayed values are even lower than this, but flow values of ~0.1 cm/s are likely to be too low, and the purpose of this analysis is to understand the limits to which you can state something accurately).

Advective flux = u * dnTA/dx, so at flow speeds of u = 0.5 and 1 cm/s, your advective flux = 1.35 and 2.7 umol/kg/hr

At the assumed low flow speed (0.5 cm/s), the "missing" advective flux is equivalent to ~ 27%, 13.5%, and 9% of your respective estimated dnTA/dt

At the assumed high flow speed (1 cm/s), the "missing" advective flux is equivalent to ~54%, 27%, and 18% of your respective estimated dnTA/dt

These values of u * dnTA/dx, relative to dnTA/dt, are sufficiently large such that they cannot be ignored (i.e. they are not ~1% or even 5% of your estimates; they may be as high at 50%).

As I described before, I believe your estimates of dnTA/dt, and hence NEC, are actually equal to dnTA/dt + u * dnTA/dx. If you knew the directionality of the flow (the sign of u), you calculate whether the term is equal to dnTA/dt + u * dnTA/dx (when u>0) or dnTA/dt - u * dnTA/dx (when u<0). In the absence of information on flow directionality, I think you have to treat the advective flux as an error on both sides of your NEC estimate (i.e. NEC +/- error). Practically speaking, this means that the lower bound on your NEC estimate becomes:

NEC_lower_bound = (rho* h) / -2 * (dnTA/dt - u * dnTA/dx)

NEC_mean = (rho * h) / -2 * dnTA/dt

NEC_upper_bound = (rho* h) / -2 * (dnTA/dt + u * dnTA/dx)

Since the spatial gradients may change throughout the day, your error bounds may as well. I will leave it up to you to choose an appropriately *conservative* value of u, recognizing that all of your recorded data are below the instrumental limits of detection.

The same set of calculations need to done for both NEP_DO and NEP_DIC. Given that TA and DIC are correlated, and similar in value, I think you could use the same value for dnDIC/dx and dnTA/dx.

And finally, the NEC error needs to be propagated through your DIC-based NEP calculations (Eq. 3), in addition to the error on the DIC fluxes. Then, the daily-integrated estimates need to have error estimates that propagate through the associated uncertainty for the hourly measurements.

I know this is difficult (none of us enter into environmental science in order to revolutionize error propagation :). I am not asking you to do this because I want to torture you. But I believe the assertions of net heterotrophy and dissolution are not very robust now, and I believe a more thorough treatment of all the errors in your calculations leading to your assertions will help you and readers assign appropriate confidence in your reported net heterotrophy and dissolution. It may even prevent someone else from rebutting your study since you are exposing all of your study's strengths and weaknesses.

**Specific comments:**
Eq. 1: This model still does not include a term for the production (consumption) of TA due to positive (negative) NEP, despite the negatively sloped line in Fig. 3 that acknowledges the relationship between DIC uptake and TA production. I had mentioned this is my previous review, but I think the comment was missed. I think the authors now

do a good job acknowledging that the simple TA and DIC models cannot resolve sulfate reduction and denitrification, but they are implicitly acknowledging the role of organic matter production in the TA budget in Fig. 3. I think Eq. 1 needs to be reformatted to include this term. Doing so, would mean that the TA budget might look something like this:

$$dnTA/dt = -2 * NEC / rho * h + 17/106 * (NEP / rho * h)$$

Thus, NEC would now look like this:

$$NEC = [(rho * h * dnTA/dt) - (16/107 * NEP)] / -2$$

Along this same line of logic, the statement that the delta_nTA/delta_nDIC slope is ~0 for ecosystem metabolism (p. 11, L7) is incorrect.

Fig. 2: I still find Fig. 2 difficult to follow. Panels a) and b) have primary y-axes that seem to indicate that PAR will be displayed in black, yet the legend in b) indicates that PAR will be shown in red and green (a color combination that is particularly problematic for colorblind individuals). Similarly, the secondary y-axis suggests that U_10 will be shown in red, but the legend indicates a times series displayed by the black lines. I still find the inclusion of four times series on a single plot, such as in panels c-h), difficult to read and to keep track of differences between the two sites. I recommend further revising this figure, possibly to display the different field sites as the different columns and to include the two sampling campaigns as different data series on each plot. After all, the one of the primary goals is to compare and contrast between the two field sites, right? Further, I recommend that the authors demo several versions of the figure with colleagues and get their feedback about the figure readability.

---

## Referee Report (RR2)

Van Dam et al. have made significant improvements to this manuscript and have addressed all of my major constructive criticisms. I am please to recommend this manuscript for publication.

There are several areas where I recommend the authors address minor outstanding issues to improve the rigor and accessibility of the manuscript. I have outlined them here for consideration:

P. 8, L 11: I understand the reasons for omitting the effects of NEP on the total alkalinity budget, but the statement "… indicated that this TA production was small compared to total NEC" would be more robust if it was stated quantitatively. For example, "… indicated that this TA production was never more than X% of the NEC."

Fig. 4: I find this figure to be nearly unreadable. Overlapping NEC, NEP_DIC, and NEP_DO on the same plots with small markers is confusing. I strongly recommend that the authors reformat this plot to add more panels such that only a single metabolic rate (NEC, NEP_DIC, or NEP_DO) is displayed per plot.

On a broader note, is Fig. 4 necessary since the same information is found in Fig. 5? In my opinion, Fig. 5 is a "truer" representation of the data collected. If you want to show composite daily cycles, I recommend re-ordering Figs. 4 and 5.

Fig. 5: I also find this figure to be largely unreadable. No reader giving any reasonable amount of effort is going to be able to understand three overlapping metabolic rate time series, each with associated error bounds which are all the same color. I strongly recommend that the authors split this figure into multiple panels such that each metabolic rate is on a different panel. An improvement would be to display the error bounds in the same color as the mean metabolic rate time series. An even better improvement could be to display the range (upper bound to lower bound) as a filled ribbon with the mean estimates as a solid line in the middle.

Fig. 6a: The overlapping error bounds for the daytime and nighttime measurements are unreadable. If the authors staggered the error bounds (move one a little to the left and the other a little the right) and used color coordination between the bar plots and error bars, it would greatly improve the ability of a reader to comprehend this figure.

---

## Author Response (AR2)

We thank reviewer 1 for once more providing thorough and constructive criticism, which we think has vastly improved this manuscript. It is clear that they have thought deeply about the shortcomings and virtues of our study, and we sincerely appreciate her/his effort in this regard. While we were at first reluctant to take the approach that reviewer 1 suggested, we have done so now, and incorporated this analysis into the manuscript. We took the conservative step of considering flow speeds of 1 cm/s, and used spatial gradients for TA and DIC discussed previously (75 µmol/kg). The spatial gradient in DO was determined similarly, but was much smaller, at 4.6 µmol/kg. These new calculations are described in the new section 2.8.

This additional uncertainty analysis does cause us to be somewhat more cautious in interpreting the net metabolism rates calculated using our 'open water' approach. However, our primary findings of net dissolution and heterotrophy have not changed. In fact, we agree with reviewer 1 that this new analysis helps us to better describe the limitations and strengths of our study in a more open manner. We have added a new figure (Fig 5) which shows a time series of metabolic rates over both sampling campaigns, including the upper and lower uncertainty bounds. These uncertainty bounds have replaced standard deviations for the error bars in figures 6 (old Fig 5) and S1. We feel that these error bounds are more appropriate and conservative than the standard deviations presented earlier, which only consider temporal variability in metabolic rates, rather than an honest estimate of real uncertainty.

Substantial additions were required to the text in section 3.2 to explain this new analysis and how it affects our interpretation of the NEP and NEC results. Shorter passages were added to the conclusion and abstract to reiterate the points laid out in section 3.2.

Other notes:

While we have taken the above steps to explore the uncertainty due to advection, we are not able to propagate this uncertainty in NEC through the NEP calculations, as also suggested by the reviewer. As stated at the end of section 2.7, the implicit consideration of $NEP_{DIC}$ into the calculation of NEC (Eq. 1) introduces an unresolvable circular reference in Eq. 3 (which includes NEC).

The reference to an expected slope of ~0 for TA vs DIC was intended to represent net productivity based on mixed NO3 and NH4, but we agree that this is confusing. The text has been changed to show a slope of -0.15, consistent with figure 3.

The inclusion of TA produced by NEP into the NEC calculations likewise introduces a circular reference, which cannot be resolved. As we mentioned in a response earlier, we do not have clear evidence to distinguish whether NEP is fueled by NH4 or NO3. Productivity driven by these different N sources would have opposite effects on TA, so we cannot comfortably say whether NEP would need to be scaled by 16/107 or -16/107. For these reasons, we feel comfortable in excluding the NEP effect on TA into NEC calculations. We hope that the additional text added to the manuscript previously, discussing alternative sources/sinks of TA, adequately addresses reviewer 1's (and 2's) concerns on this topic.

We agree that Figure 2 was hard to follow, and have substantially revised the layout in line with reviewer 1's suggestions. Red-green combinations have been removed, and plots with more than two parameters have been split into separate subplots.

Van Dam et al. have put a lot of work into addressing the three very thorough reviews of their initial submission. In general, I think they did a good job addressing my constructive criticisms of the initial manuscript and it is much improved.

I appreciated Van Dam et al.'s response to my primary constructive criticism RE: missing advective terms in their metabolic rate models. They did a nice job diving back into their data to assess spatial gradients in TA and attempted to place them into context with their diel TA ranges to argue that advective fluxes can be safely ignored, after describing in the text. Below, I will show why I disagree with the assertion that these advective fluxes can be ignored. I will walk the authors through some examples and explain why I believe the authors should use these "missing" advective fluxes as error bounds on their metabolic rates.

In this example, we will continue to stick to NEC and TA fluxes, but the same concepts are analogous for DIC (and will need to be applied equally).

As the authors stated int their response, they observed spatial TA gradients of 75 umol/ kg/km. Let's call this variable: dnTA/dx.

Now, observing NEC rates of +/- 15 mmol/m^2/hr (Fig. 6c), we can invert Eq. 1 to calculate the time rate of change of TA (dnTA/dt). Let's assume rho=1025 kg/m^3 and h = 2 meters. Furthermore, let's also do the calculation for NEC rates of 5, 10, and 15 mmol/m^2/hr (we will be sign-agnostic because it's the magnitudes we are most interested in).

So inverting Eq. 1 to solve for the dnTA/dt yields: dnTA/dt = -2 * NEC / rho * h

And so for our three test cases of NEC = 5, 10, and 15 mmol/m^2/hr, dnTA/dt ~ 5, 10, and 15 umol/kg/hr (with some simplified rounding)

Now let's compare these estimates for the time-varying term against your spatial gradients. Your plots of TCMs suggest flow speeds below 1 cm/s (acknowledging that the limit of detection on the instrument is 2 cm/s). So let's consider two test cases of u= 0.5 cm/s and u = 1cm/s (I know the displayed values are even lower than this, but flow values of ~0.1 cm/s are likely to be too low, and the purpose of this analysis is to understand the limits to which you can state something accurately).

Advective flux = u * dnTA/dx, so at flow speeds of u = 0.5 and 1 cm/s, your advective flux = 1.35 and 2.7 umol/kg/hr

At the assumed low flow speed (0.5 cm/s), the "missing" advective flux is equivalent to ~ 27%, 13.5%, and 9% of your respective estimated dnTA/dt

At the assumed high flow speed (1 cm/s), the "missing" advective flux is equivalent to ~54%, 27%, and 18% of your respective estimated dnTA/dt

These values of u * dnTA/dx, relative to dnTA/dt, are sufficiently large such that they cannot be ignored (i.e. they are not ~1% or even 5% of your estimates; they may be as high at 50%).

As I described before, I believe your estimates of dnTA/dt, and hence NEC, are actually equal to dnTA/dt + u * dnTA/dx. If you knew the directionality of the flow (the sign of u), you calculate whether the term is equal to dnTA/dt + u * dnTA/dx (when u>0) or dnTA/dt - u * dnTA/dx (when u<0). In the absence of information on flow directionality, I think you have to treat the advective flux as an error on both sides of your NEC estimate (i.e. NEC +/- error). Practically speaking, this means that the lower bound on your NEC estimate becomes:

NEC_lower_bound = (rho* h) / -2 * (dnTA/dt - u * dnTA/dx) NEC_mean = (rho * h) / -2 * dnTA/dt NEC_upper_bound = (rho* h) / -2 * (dnTA/dt + u * dnTA/dx)

Since the spatial gradients may change throughout the day, your error bounds may as well. I will leave it up to you to choose an appropriately *conservative* value of u, recognizing that all of your recorded data are below the instrumental limits of detection.

The same set of calculations need to done for both NEP_DO and NEP_DIC. Given that TA and DIC are correlated, and similar in value, I think you could use the same value for dnDIC/dx and dnTA/dx.

And finally, the NEC error needs to be propagated through your DIC-based NEP calculations (Eq. 3), in addition to the error on the DIC fluxes. Then, the daily-integrated estimates need to have error estimates that propagate through the associated uncertainty for the hourly measurements.

I know this is difficult (none of us enter into environmental science in order to revolutionize error propagation :). I am not asking you to do this because I want to torture you. But I believe the assertions of net heterotrophy and dissolution are not very robust now, and I believe a more thorough treatment of all the errors in your calculations leading to your assertions will help you and readers assign appropriate confidence in your reported net heterotrophy and dissolution. It may even prevent someone else from rebutting your study since you are exposing all of your study's strengths and weaknesses.

**Specific comments:**

Eq. 1: This model still does not include a term for the production (consumption) of TA due to positive (negative) NEP, despite the negatively sloped line in Fig. 3 that acknowledges the relationship between DIC uptake and TA production. I had mentioned this is my previous review, but I think the comment was missed. I think the authors now

do a good job acknowledging that the simple TA and DIC models cannot resolve sulfate reduction and denitrification, but they are implicitly acknowledging the role of organic matter production in the TA budget in Fig. 3. I think Eq. 1 needs to be reformatted to include this term. Doing so, would mean that the TA budget might look something like this:

dnTA/dt = -2 * NEC / rho * h + 17/106 * (NEP / rho * h)

Thus, NEC would now look like this:

NEC = [(rho * h * dnTA/dt) - (16/107 * NEP)] / -2

Along this same line of logic, the statement that the delta_nTA/delta_nDIC slope is ~0 for ecosystem metabolism (p. 11, L7) is incorrect.

Fig. 2: I still find Fig. 2 difficult to follow. Panels a) and b) have primary y-axes that seem to indicate that PAR will be displayed in black, yet the legend in b) indicates that PAR will be shown in red and green (a color combination that is particularly problematic for colorblind individuals). Similarly, the secondary y-axis suggests that U_10 will be shown in red, but the legend indicates a times series displayed by the black lines. I still find the inclusion of four times series on a single plot, such as in panels c-h), difficult to read and to keep track of differences between the two sites. I recommend further revising this figure, possibly to display the different field sites as the different columns and to include the two sampling campaigns as different data series on each plot. After all, the one of the primary goals is to compare and contrast between the two field sites, right? Further, I recommend that the authors demo several versions of the figure with colleagues and get their feedback about the figure readability.

[revised manuscript text omitted]

---

## Author Response (AR3)

Van Dam et al. have made significant improvements to this manuscript and have addressed all of my major constructive criticisms. I am please to recommend this manuscript for publication. There are several areas where I recommend the authors address minor outstanding issues to improve the rigor and accessibility of the manuscript. I have outlined them here for consideration:

P. 8, L 11: I understand the reasons for omitting the effects of NEP on the total alkalinity budget, but the statement "... indicated that this TA production was small compared to total NEC" would be more robust if it was stated quantitatively. For example, "... indicated that this TA production was never more than X% of the NEC."

We have added an average difference between the two approaches to L11.

Fig. 4: I find this figure to be nearly unreadable. Overlapping NEC, NEP_DIC, and NEP_DO on the same plots with small markers is confusing. I strongly recommend that the authors reformat this plot to add more panels such that only a single metabolic rate (NEC, NEP_DIC, or NEP_DO) is displayed per plot.

On a broader note, is Fig. 4 necessary since the same information is found in Fig. 5? In my opinion, Fig. 5 is a "truer" representation of the data collected. If you want to show composite daily cycles, I recommend re-ordering Figs. 4 and 5.

Figure 4 was overhauled as per the reviewer's suggestion. We now show each metabolic rate in its own subplot, separated into columns by site. This re-formatting significantly improved the readability of the figure, which we would like to keep in the manuscript, despite the overlap with figure 5.

Fig. 5: I also find this figure to be largely unreadable. No reader giving any reasonable amount of effort is going to be able to understand three overlapping metabolic rate time series, each with associated error bounds which are all the same color. I strongly recommend that the authors split this figure into multiple panels such that each metabolic rate is on a different panel. An improvement would be to display the error bounds in the same color as the mean metabolic rate time series. An even better improvement could be to display the range (upper bound to lower bound) as a filled ribbon with the mean estimates as a solid line in the middle.

We opted to take the reviewer's first suggestion and the error bounds are now represented in the same color as the mean metabolic rate. We have also made the figure larger, which should improve the readability.

Fig. 6a: The overlapping error bounds for the daytime and nighttime measurements are unreadable. If the authors staggered the error bounds (move one a little to the left and the other a little the right) and used color coordination between the bar plots and error bars, it would greatly improve the ability of a reader to comprehend this figure.

Figure 6 was also updated per the reviewer's suggestion. We have staggered the bars to help with the interpretation of overlapping error bars. We have also fixed the colors in figure 6b, which were problematic for those with red-green colorblindness.

[revised manuscript text omitted]